# Learning Multi-Timescale Abstractions for Hierarchical Combinatorial Planning

Vivienne Huiling Wang [1]  Tinghuai Wang [2]  Joni Pajarinen [1]

## Abstract

The combination of exponentially large action spaces, stochastic dynamics, and long-horizon decision-making under limited resources makes Sequential Stochastic Combinatorial Optimization (SSCO) particularly challenging for reinforcement learning. Hierarchical Reinforcement Learning (HRL) offers a natural decomposition, but it places the high-level policy in a Semi-Markov Decision Process (SMDP) where actions have variable durations, making it difficult to learn a world model that is suitable for planning. We introduce a model-based hierarchical framework for sequential stochastic combinatorial decision-making that directly addresses this issue. Our method combines a latent-space tree-search planner with an SMDP-aware world model for variable-duration decisions. A multi-timescale objective structures the latent dynamics so that transition magnitudes reflect the effective temporal scales of abstract actions, enabling efficient lookahead under adaptive temporal abstraction. We further learn a subgoal-conditioned budget policy jointly with the world model to support context-aware resource allocation. Across challenging SSCO benchmarks, our method outperforms strong baselines.

## 1. Introduction

Sequential Stochastic Combinatorial Optimization (SSCO) problems require making a sequence of combinatorial decisions under uncertainty in order to maximize a cumulative objective (Li et al., 2018; Kool et al., 2019). They arise in applications such as dynamic vehicle routing (Kool et al., 2019) and adaptive influence maximization in social networks (Chen et al., 2021). SSCO is particularly challenging because the action space is combinatorial, the dynamics are

[1]Department of Electrical Engineering and Automation, Aalto University, Finland [2]Qutwo, Finland. Correspondence to: Vivienne Huiling Wang <vivienne.wang@aalto.fi>.

*Proceedings of the 43rd International Conference on Machine Learning*, Seoul, South Korea. PMLR 306, 2026. Copyright 2026 by the author(s).

stochastic and path-dependent, and early decisions can have long-lasting consequences. For instance, in Adaptive Influence Maximization (AIM), an agent repeatedly selects seed nodes to induce stochastic cascades, while in the Stochastic Orienteering Problem (SOP), an agent plans multi-day routes on a graph subject to tight travel budgets. In both settings, strategic aims such as "target a specific community" or "sweep a profitable region" can require very different amounts of effort to execute.

Flat reinforcement learning (RL) methods often struggle in these settings: they must act directly in an enormous action space and solve long-horizon credit assignment without explicit structure. Hierarchical RL (HRL) (Dayan & Hinton, 1992; Barto & Mahadevan, 2003; Sutton et al., 1999) offers a natural way to introduce temporal abstraction, but in SSCO it comes with a structural twist. At the high level, the agent must decide not only *what* to do (subgoal) but also *how much resource* to spend (budget). Each high-level decision specifies a subgoal together with a budget, and the low level then acts until either the subgoal terminates or the budget is exhausted. This induces variable numbers of primitive steps between high-level decisions, so the high-level process is a *Semi-Markov Decision Process (SMDP)* (Sutton et al., 1999) rather than a standard one-step MDP.

This SMDP structure clashes with many existing goal-conditioned HRL and multi-timescale world-model methods. Approaches such as Director (Hafner et al., 2022) and MTS-WM (Shaj Kumar et al., 2023) impose a fixed slow timescale: the high level acts every fixed number of primitive steps, and the world model predicts at that fixed stride. This effectively turns the high level back into an MDP with fixed-duration macro steps. In SSCO, however, the agent must choose *how long* to pursue each subgoal, depending on both the state and the goal. Simply fixing the stride cannot express this. The closest SSCO baseline, WS-option (Feng et al., 2025), supports variable budgets but communicates only a scalar allocation: it decides *how much* resource to spend without specifying *what* strategic subtask to pursue. As a result, the low-level controller receives no semantic guidance and may under-spend (failing to complete an intended subtask) or over-spend (wasting scarce budget on low-value progress). Moreover, WS-option's high level is model-free, so it is inherently limited in explicit lookahead to anticipate how stochastic, path-dependent SSCO dynam-

ics will evolve under variable-duration budgeted decisions.

We introduce Learning *Multi-Timescale Abstractions (LMTA)*, a model-based, goal-conditioned HRL framework designed around this SMDP structure. At the high level, LMTA uses a latent-space model-based tree search planner (Schrittwieser et al., 2020) to search over a set of learned subgoals, while a learned budget head chooses how much resource to allocate to each selected subgoal. To support this planner, we learn an *SMDP-aware world model* that maps a state and a subgoal directly to a post-subgoal latent state and a cumulative reward in a single macro step. Rather than predicting durations explicitly, which can be brittle, we propose a *Multi-Timescale SMDP (MTS-SMDP) world model* in which temporal scale is encoded in the latent geometry: the magnitude of a macro transition reflects how long the corresponding subgoal tends to run. The same latent space also supports low-level, per-step dynamics.

Our main contributions are:

- **Goal-conditioned hierarchical planning in an SMDP.** We formulate the high level of SSCO as an SMDP where each decision selects a learned subgoal together with a budget. LMTA uses a latent-space model-based tree search planner to search over these subgoals, enabling model-based lookahead over variable-duration abstract actions rather than fixed-stride options or budget-only signals.
- **Multi-timescale SMDP world-model learning via latent geometry.** We propose a unified objective that shapes a single latent space to reflect both micro (primitive-step) and macro (subgoal-level) dynamics. The objective balances complementary constraints that jointly structure the latent transitions, so that learned latent displacements correlate with the effective durations of subgoals. We show that this geometry–duration relationship emerges from the objective and relate it to planning regret.
- **Subgoal-conditioned budgets and SSCO performance.** We couple the world model with a subgoal-conditioned budget head, so that the high level decides what strategy to pursue and how much resource to invest in it. On large-scale AIM, SOP, and a Power-2500 benchmark, LMTA outperforms strong model-free and model-based baselines, highlighting the value of SMDP-aware, multi-timescale planning for stochastic combinatorial optimization.

**Conflict of Interest Disclosure.** The authors declare no financial conflicts of interest related to this work. This study evaluates public benchmarks and does not evaluate models, products, or services developed by the authors' respective employers.

## 2. Related Works

**Hierarchical Reinforcement Learning.** HRL methods tackle long-horizon problems by creating temporal abstractions (Barto & Mahadevan, 2003). The options framework (Sutton et al., 1999) and derivatives like Option-Critic (Bacon et al., 2017) formalize temporally extended actions. Recent work learns subgoals either as explicit states with hindsight relabeling (Nachum et al., 2018; Levy et al., 2019; Wang et al., 2023), through diffusion-based subgoal generation (Wang et al., 2025), or as learnable continuous representations (Li et al., 2021; Wang et al., 2024). While effective, these are typically *model-free* at the high level, limiting explicit forward planning over SMDP dynamics under stochasticity. WS-option (Feng et al., 2025) is the closest SSCO baseline: it is model-free at the high level (no long-term model-based planning), does not explicitly model SMDP dynamics, and emits only a scalar budget, offering limited semantic guidance to the low level.

**Hierarchical Planning and Search.** Prior work has combined MCTS with temporal abstraction to tackle long horizons. Approaches like Hierarchical MCTS (Vien & Toussaint, 2015) and Option-MCTS (Bai et al., 2016) extend tree search to operate over options, effectively reducing the search depth. However, these methods typically rely on pre-defined options or do not explicitly model the variable-duration dynamics required for resource-constrained SSCO. In the context of subgoal generation, methods like LEAP (Nasiriany et al., 2019) perform search over subgoals to guide a low-level policy. Unlike these approaches, which often assume fixed representations or distance metrics, LMTA integrates a latent-space tree search planner with a learned multi-timescale world model, where the latent geometry itself adapts to the temporal scale of the subgoals.

**Modeling for Semi-Markov Decision Processes.** A key challenge is that the high level operates in an SMDP with variable action durations. Traditional MDP world models assume fixed-duration transitions. Prior work models option duration explicitly via termination learning (Bacon et al., 2017; Khetarpal et al., 2020) or separate duration predictors (Harb et al., 2018), which can be difficult to train and may not capture the interplay between a subgoal's semantics and its execution time. Many goal-conditioned HRL approaches instead assume fixed-length temporal abstractions (Nachum et al., 2018; Sharma et al., 2019; Zhang et al., 2021), simplifying planning but sacrificing flexibility. Our approach learns temporal scale implicitly through latent geometry.

**Unified SMDP Dynamics and Model-based Planning.** Neural ODEs model continuous-time dynamics over arbitrary intervals (Chen et al., 2018), but require costly solvers inside planning loops and do not naturally yield discrete, temporally abstract actions central to HRL. Model-based RL often learns one-step latent dynamics (*e.g.*, Dreamer

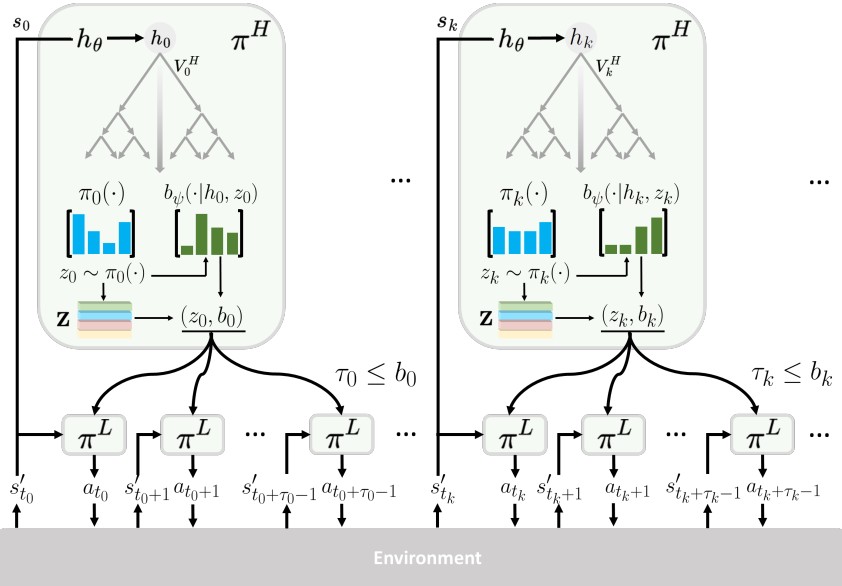

*Figure 1.* Architecture of LMTA. At each high-level (HL) step, the planner encodes the state as $h_k$ and runs latent-space tree search over subgoals to produce a search policy $\pi_k(\cdot)$. A subgoal $z_k$ is selected, after which the budget head samples $b_k$, yielding HL action $(z_k, b_k)$. The low-level (LL) policy $\pi^L$ executes primitive actions for $\tau_k$ steps to pursue $z_k$, where $\tau_k \leq b_k$.

(Hafner et al., 2020)); when extended hierarchically (Director (Hafner et al., 2022)), the high level sets latent goals, yet the underlying world model remains one-step. Evaluating a single high-level goal thus requires multi-step latent rollouts of the low-level policy within each node expansion of the search tree. In contrast, our dynamics function $g_\theta$ is trained to predict the *post-subgoal* latent in a single step, collapsing a variable-duration abstract action into one model evaluation. Our MTS-SMDP framework structures the latent space so that transition magnitudes correlate with temporal scale, enabling SMDP-aware planning without an explicit duration predictor. MTS-WM (Shaj Kumar et al., 2023) also models multiple timescales but uses a fixed hyperparameter $H$ to update the slow latent, defining a fixed temporal stride. In contrast, our framework handles agent-chosen, variable durations inherent to SSCO.

**RL for Combinatorial Optimization.** Applying RL to CO has gained traction (Mazyavkina et al., 2021). Seminal works address TSP (Bello et al., 2017) and VRP (Kool et al., 2019) using Pointer Networks (Vinyals et al., 2015) and Transformers, typically learning one-pass constructive policies for static CO. LMTA targets the more complex SSCO setting with stochastic transitions and multi-stage resource planning, where hierarchy is essential. Within SSCO, WS-option (Feng et al., 2025) provides an option-based baseline; in contrast, LMTA enables model-based, goal-conditioned planning with an SMDP-aware world model and subgoal-conditioned budget allocation. SSCO challenges general RL because actions are combinatorial, rewards are non-myopic, and intertemporal budget constraints

require variable-duration planning, which fixed-stride HRL methods like Director cannot handle efficiently.

## 3. Preliminaries

**Markov and Semi-Markov Decision Processes** A standard Markov Decision Process (MDP) is defined by a tuple $\mathcal{M} = (\mathcal{S}, \mathcal{A}, \mathcal{P}, \mathcal{R}, \gamma_{\mathrm{LL}})$, where $\mathcal{S}$ is the state space, $\mathcal{A}$ is the action space, $\mathcal{P} : \mathcal{S} \times \mathcal{A} \to \Delta(\mathcal{S})$ is the transition probability function, $\mathcal{R} : \mathcal{S} \times \mathcal{A} \to \mathbb{R}$ is the reward function, and $\gamma_{\mathrm{LL}} \in [0, 1)$ is the *primitive-time* discount factor. The goal is to find a policy $\pi : \mathcal{S} \to \Delta(\mathcal{A})$ that maximizes the expected cumulative discounted reward $V^\pi(s) = \mathbb{E}_\pi[\sum_{t=0}^\infty \gamma_{\mathrm{LL}}^t \mathcal{R}(s_t, a_t) \mid s_0 = s]$.

Hierarchical reinforcement learning often induces a Semi-Markov Decision Process (SMDP) at the high level. In an SMDP, actions (often called options or subgoals) can take a variable number of primitive timesteps to complete. An SMDP is defined by a tuple $(\mathcal{S}, \mathcal{A}_{HL}, \mathcal{P}_\tau, \mathcal{R}_\tau, \gamma)$, where $\mathcal{A}_{HL}$ is the set of high-level actions and $\gamma \in [0, 1)$ is a *decision-time* discount applied per high-level decision. When a high-level action $a^H \in \mathcal{A}_{HL}$ is taken in state $s$, it executes for a stochastic duration $\tau$, drawn from a distribution $p(\tau \mid s, a^H)$. The system transitions to a new state $s'$ according to $\mathcal{P}_\tau(s' \mid s, a^H)$, and a cumulative *primitive-time* discounted reward is received: $R_\tau(s, a^H) = \mathbb{E}\left[\sum_{i=0}^{\tau-1} \gamma_{\mathrm{LL}}^i r_{t+i}\right]$, where $r_{t+i}$ denotes the primitive reward during the execution of $a^H$. The *primitive-step-equivalent* SMDP Bellman backup would discount the continuation by $\gamma_{\mathrm{LL}}^\tau$, i.e., $V(s) = \mathbb{E}\left[R_\tau(s, a^H) + \gamma_{\mathrm{LL}}^\tau V(s')\right]$,

which depends on the random duration $\tau$. In our setting, for planning/backups we instead treat each high-level decision as one step with a fixed decision-time discount $\gamma$; this is an *algorithmic approximation* that simplifies tree-search backups and stabilizes optimization, but induces an effective time preference defined per decision epoch rather than strict primitive-step equivalence. The key challenge remains that both the transition dynamics and rewards depend on the variable duration $\tau$.

In the SSCO setting, the high level chooses a **subgoal** $z$ and a **budget** $b$. The low level then acts for a variable number of primitive steps (up to $b$), inducing a random duration $\tau$ between high-level decisions. Unlike standard HRL which often enforces a fixed high-level stride, SSCO requires allocating varying budgets, making the time between decisions dynamic. This invalidates fixed-stride assumptions of methods like Director (Hafner et al., 2022) and motivates an SMDP formulation.

**Model-Based Planning with Latent-Space Tree Search**
We consider model-based planning methods that combine tree search with a learned dynamics model. In particular, we follow the latent-space tree search paradigm of MuZero (Schrittwieser et al., 2020), which performs Monte Carlo Tree Search (MCTS) using learned representation, dynamics, and prediction functions:

- $h_\theta : \mathcal{S} \to \mathcal{H}$ maps an observation to a latent state $h \in \mathcal{H}$.
- $g_\theta : \mathcal{H} \times \mathcal{A} \to \mathcal{H} \times \mathcal{R}$ predicts the next latent state and macro reward. As in MuZero, the reward prediction is produced by the dynamics function, *i.e.*, $g_\theta$ outputs both $(h', R)$.
- $f_\theta : \mathcal{H} \to \Delta(\mathcal{A}) \times \mathcal{R}$ predicts a policy prior $\mathbf{p}$ and value $v$.

Tree search in the latent space produces an improved policy target, and we adopt this planning paradigm for our high-level controller.

## 4. Method

In order to address the challenges of exponentially large combinatorial action spaces, stochastic and path-dependent dynamics, and long-horizon decision-making under intertemporal resource constraints, we propose LMTA, a hierarchical RL framework for sequential stochastic combinatorial decision-making. A key modeling requirement in this setting is to support high-level decisions that induce variable numbers of primitive steps, so planning must operate over SMDP transitions rather than fixed-stride macro steps. LMTA meets this requirement with three coupled components: (1) a high-level latent-space model-based tree search planner, (2) a unified multi-timescale SMDP world model that supports planning over variable-duration subgoals, and (3) a subgoal-conditioned budget policy that allocates re-

sources across decision stages.

### 4.1. High-Level Planning with Model-Based Search

A central component of our framework is a latent-space model-based tree search planner following MuZero (Schrittwieser et al., 2020) at the high level, which enables lookahead search over abstract subgoals. We provide network architecture details in Appendix F. This design choice is motivated by the unique demands of SSCO. Unlike traditional control problems where actions have immediate, localized effects, decisions in SSCO have long-lasting consequences under uncertainty. A model-based planner allows the agent to simulate potential sequences of subgoals and their likely outcomes, which is critical for effective resource allocation and strategic positioning.

**High-level SMDP.** We index primitive (low-level) interaction by $t = 0, 1, \ldots$ with states $s'_t$. We index high-level decisions by $k = 0, 1, \ldots$. At high-level decision step $k$, the agent observes the high-level state $s_k$, selects a subgoal $z_k \in \mathcal{Z}$ and a budget $b_k \in \{0, \ldots, B_k\}$ (where $B_k$ is the remaining resource), and the low-level controller then executes primitive actions for a random duration $\tau_k \in \{0, \ldots, b_k\}$, determined by subgoal termination or budget exhaustion. During this execution, primitive transitions follow $s'_t \to s'_{t+1}$ for $t = t_k, \ldots, t_k + \tau_k - 1$, where $t_k$ denotes the primitive time at which the $k$-th high-level decision is taken (with $t_0 = 0$). The next high-level decision occurs at primitive time $t_{k+1} := t_k + \tau_k$, and the corresponding next high-level state is $s_{k+1}$. The cumulative *primitive-time* discounted reward attributed to the high-level action $a_k^H = (z_k, b_k)$ is $R_k = \sum_{i=0}^{\tau_k - 1} \gamma_{\text{LL}}^i r_{t_k+i}$. This induces a semi-Markov transition $s_k \xrightarrow[(\tau_k, R_k)]{a_k^H} s_{k+1}$ with variable duration $\tau_k$ and reward $R_k$. For tree-search backups, we use a *decision-time* discount factor $\gamma \in [0, 1)$ applied per high-level decision (we separate notation from $\gamma_{\text{LL}}$ to avoid confusion).

The high-level planner consists of three learned components, each tailored for SSCO:

1. **Representation function $h_\theta : \mathcal{S} \to \mathcal{H}$**: This function maps the high-dimensional, often graph-structured state $s$ of an SSCO problem into a compact latent representation $h$. For AIM and SOP, we use Graph Neural Networks (GNNs) that can effectively capture the topological structure and node features, producing a fixed-size vector that summarizes the global state of the system.
2. **Dynamics function $g_\theta : \mathcal{H} \times \mathcal{Z} \to (\mathcal{H}, \mathcal{R})$**: This function predicts the next *decision-time* latent state $h_{k+1}$ (the latent component is denoted by $g_\theta^h(h, z)$) and the cumulative reward $R_k \in \mathcal{R}$ resulting from executing a high-level subgoal $z_k$. Since our high-level planner branches only over $z$ (for tractable search in SSCO), the budget

$b_k$ is selected *after* planning by a subgoal- and state-conditioned policy $b_\psi(\cdot \mid h_k, z_k)$. Accordingly, $g_\theta$ is trained to model the *budget-marginalized* SMDP transition induced by the learned budget rule and low-level controller:

$$\mathcal{P}^{\pi^L, b_\psi}(s_{k+1}, R_k \mid s_k, z_k) \triangleq \mathbb{E}_{b_k \sim b_\psi(\cdot \mid h_k, z_k)}$$
$$\left[\mathcal{P}(s_{k+1}, R_k \mid s_k, (z_k, b_k), \pi^L)\right], \quad (1)$$

so that $g_\theta^h(h_k, z_k)$ approximates the post-subgoal latent under the endogenous distribution $b_\psi(\cdot \mid h_k, z_k)$, and $R_\theta(s_k, z_k)$ approximates the corresponding macro-return. [1] The abstract subgoals are represented as learnable embeddings, allowing their semantic meaning to be discovered end-to-end during training. The subgoal set $\mathcal{Z} = \{z_1, \dots, z_M\}$ is a finite learned dictionary of embeddings, not a geometric subspace of the latent state space $\mathcal{H}$.

3. **Prediction function** $f_\theta : \mathcal{H} \to \mathcal{P} \times \mathcal{V}$: From a latent state $h_k$, this function outputs:
   - A **policy prior** $\mathbf{p}_k \in \mathcal{P}$ over subgoals $\mathcal{Z}$.
   - A **value** $v_k \in \mathcal{V}$ estimating the expected future return from $h_k$, where $\mathcal{V} \subset \mathbb{R}$.

   These predictions guide MCTS, focusing planning on promising subgoals and enabling efficient evaluation without rolling out to episode end.

**Planning process.** The HL action is $a_k^H = (z_k, b_k)$ with $\mathcal{A}_{HL} = \mathcal{Z} \times \mathcal{B}$. The planner operates as follows:

1. Encode the current HL state: $h_k = h_\theta(s_k)$.
2. Run MCTS over subgoals $z \in \mathcal{Z}$ using the learned latent dynamics and prediction models ($g_\theta$ and $f_\theta$).
3. Obtain the search policy $\pi_k$ (from visit counts) and select $z_k \sim \pi_k$.
4. Sample the budget from the subgoal-conditioned budget policy: $b_k \sim b_\psi(\cdot \mid h_k, z_k)$.

### 4.2. Unified Multi-Timescale World Model Learning

A central difficulty in our setting is that the high-level policy lives in an SMDP: executing a subgoal and its budget from a state leads to a stochastic, variable number of primitive steps before the next high-level decision. We would like a *single* world model that supports both (i) micro-scale MDP dynamics at the level of primitive steps and (ii) macro-scale

dynamics at the level of subgoals that may run for many steps. In addition, this model should be *planning ready*: given a latent state and a candidate subgoal, the planner should be able to predict the post-subgoal outcome in one model evaluation, without rolling out the low-level policy inside the search. At inference time, this SMDP transition model is used for high-level search only. The low-level controller does not perform online model rollouts; instead, it benefits indirectly from the shared encoder, the learned latent geometry, and the selected subgoal embedding.

Our approach is to learn a *unified latent space* in which temporal scale is encoded geometrically. The dynamics network ($g_\theta : \mathcal{H} \times \mathcal{Z} \to \mathcal{H} \times \mathcal{R}$) implements a one-step SMDP transition $g_\theta(h_\theta(s), z) = \left(g_\theta^h(h_\theta(s), z), R_\theta(s, z)\right)$, where $g_\theta^h(h_\theta(s), z)$ summarizes the post-subgoal latent state and $R_\theta(s, z)$ is the predicted cumulative reward over the (random) subgoal duration. We define the latent distance of a macro transition as $d_\theta(s, z) \triangleq d\left(h_\theta(s), g_\theta^h(h_\theta(s), z)\right)$, where $d(\cdot, \cdot)$ is the chordal distance between unit-normalized latents on the sphere. Since all latents are $\ell_2$-normalized, we use the equivalent form $d(u, v) = \|u - v\|_2$ for clarity. We score a subgoal by $\sigma_{\theta,\phi}(s, z) \triangleq d_\theta(s, z) + m_\phi(z)$, with $m_\phi(z)$ a learned margin head that is regularized to avoid unbounded growth. In this design, the *size* of the macro transition encodes the temporal scale associated with $z$, while the shared encoder $h_\theta$ also supports per-step micro dynamics.

We structure the training objective around three forces, inspired by attractive and repulsive dynamics in contrastive learning ([Hadsell et al., 2006](#)):

1. **Micro-scale Pull.** A *low-level cap* penalizes latent jumps between consecutive primitive states that exceed a small threshold, so that one primitive step corresponds to a bounded latent distance.
2. **Macro-scale Push.** A *high-level margin* enforces that the latent displacement produced by a macro transition for subgoal $z$ is at least a learned margin $m_\phi(z)$, preventing long-duration subgoals from collapsing to negligible moves.
3. **Temporal Order.** A budget-aware ranking loss aligns the score $\sigma_{\theta,\phi}(s, z)$ with the effective duration of executed subgoals.

Together, these terms shape a multi-timescale geometry: micro dynamics are locally smooth, while macro transitions reflect variable, agent-chosen temporal scopes.

**Formal objective.** Let $\tau(s, z, b) \in \{0, \dots, b\}$ denote the realized number of primitive steps when executing the HL action $(z, b)$ from decision-time state $s$ (termination or budget exhaustion).[2] To avoid requiring paired rollouts from

---

[1] Conditioning $g_\theta$ directly on $b$ is possible, but would increase the tree-search branching factor from $|\mathcal{Z}|$ to $|\mathcal{Z}| \cdot |\mathcal{B}|$ (substantially increasing MCTS cost) in SSCO where budgets are discrete and state-dependent. Crucially, budget is *not ignored*: it is chosen as a second-stage decision conditioned on the planned work and context via $b_\psi(\cdot \mid h_k, z_k)$. Thus, from the planner's perspective, choosing $z$ induces a stochastic macro-transition that is *marginalized* over $b$; any remaining variability is treated as part of the SMDP stochasticity and is captured by the modeling-error term in Appendix A.

[2] This is the stopping time of the realized SMDP transition induced by the current policy and legality constraints, not an un-

the same decision point, we form supervision from *executed* HL transitions sampled from the experience distribution. Concretely, let $(s_i, z_i, b_i)$ and $(s_j, z_j, b_j)$ be two HL transitions drawn (typically independently) from the replay distribution, with realized durations $\tau_i := \tau(s_i, z_i, b_i)$ and $\tau_j := \tau(s_j, z_j, b_j)$. We define the pairwise label $Y \triangleq \text{sign}(\tau_i - \tau_j) \in \{-1, 0, 1\}$. We optimize three complementary losses that encourage (i) low-level MDP consistency, (ii) high-level under-displacement, and (iii) duration-aware ordering, respectively. We use $(x)_+ \triangleq \max\{0, x\}$, and let $\kappa > 0$ denote the low-level cap threshold that upper-bounds the per-step latent displacement.

$$\mathcal{L}_{\text{cap}}(\theta) := \underbrace{\mathbb{E}_{(s'_t, s'_{t+1})}\big[\big(d(h_\theta(s'_t), h_\theta(s'_{t+1})) - \kappa\big)^2_+\big]}_{\text{Low-Level MDP Consistency (\textit{Pull})}},$$

(2a)

$$\mathcal{L}_{\text{push}}(\theta, \phi) := \underbrace{\mathbb{E}_{(s, z)}\big[\big(m_\phi(z) - d_\theta(s, z)\big)_+\big]}_{\text{High-Level Under-Displacement (\textit{Push})}},$$

$$\mathcal{L}_{\text{order}}(\theta, \phi) := \underbrace{\mathbb{E}_{(s_i, z_i, b_i; s_j, z_j, b_j)}\big[\big(1 - Y\,\Delta\sigma\big)_+\big]}_{\text{Budget-Aware Order Consistency}},$$

$$\Delta\sigma := \sigma_{\theta,\phi}(s_i, z_i) - \sigma_{\theta,\phi}(s_j, z_j).$$

Here, $\mathcal{L}_{\text{cap}}$ is computed on primitive transitions $(s'_t, s'_{t+1})$ from the LL replay buffer, while $\mathcal{L}_{\text{push}}$ and $\mathcal{L}_{\text{order}}$ use HL states and executed subgoals collected during self-play. For clarity, we defer the full definition of unified objective to Equation (3) in Appendix A.1.

This unified objective is designed to impose a meaningful temporal structure on the latent space. The **Low-Level MDP Consistency** term acts as a geometric regularizer, controlling the expected per-step latent motion. As derived in **Lemma A.9** (Appendix A), minimizing this loss allows us to upper-bound the expected total latent path length of a low-level trajectory by a quantity proportional to its expected duration $\mathbb{E}[\tau]$. This establishes the foundational link between geometry and time.

The key insight is how these geometric constraints force temporal awareness to emerge. The **High-Level Under-Displacement** term encourages the latent displacement $d_\theta(s, z)$ to meet or exceed the learned margin $m_\phi(z)$. By combining this lower bound on displacement with the upper bound from the cap, **Proposition A.10** proves that $m_\phi(z)$ becomes a lower bound for the subgoal's expected duration $\mathbb{E}[\tau \mid s, z]$ under mild conditions. The planner can therefore use this geometric quantity as a learnable proxy for temporal cost, without introducing an explicit duration predictor in the world model.

Finally, the **Budget-Aware Order Consistency** term calibrates this structure for decision-making. We prove in

constrained completion time.

**Theorem A.11** that minimizing this ranking loss yields a scorer $\sigma_{\theta,\phi}$ that orders subgoals consistently with their effective, budget-aware mean durations. The full theoretical analysis in Appendix A culminates in a regret bound (**Theorem A.13**) connecting model quality to planning performance over macro-actions.

**Relation to prior HRL and multi-timescale models.** This unified objective yields what we call a *Multi-Timescale SMDP world model*: a single latent space simultaneously respects micro-scale MDP dynamics and macro-scale SMDP dynamics, with temporal scale emerging from geometry rather than from a fixed slow stride. This contrasts with:

- **Model-free options and budget-only HRL** (*e.g.*, WS-option), which primarily modulate the *extent* of execution (termination/budget) but provide no explicit high-level intent. Consequently, the low level is not directed toward a concrete *subtask* (*what* to pursue), and the high level lacks an explicit transition model for reasoning about stochastic, path-dependent SSCO evolution under variable-duration decisions.
- **Fixed-stride hierarchical world models** (*e.g.*, Director, MTS-WM), where slow dynamics are defined at a fixed interval, so the high level operates with fixed-duration macro steps. In contrast, our macro transitions compress a variable number of micro steps into one latent move, and their length adapts to the subgoal and environment.

In the appendix, we show that under mild assumptions, this objective suffices to derive a geometry-duration coupling, a duration lower bound from the learned margins, and an order-consistent scorer, and we relate these properties to planning regret (Theorems A.9 to A.11 and A.13). Empirically, we find that this multi-timescale geometry is essential for making latent-space tree search planning effective in SSCO.

### 4.3. Subgoal-Conditioned Budget Allocation

SSCO problems involve budget constraints that must be allocated across sequential decisions. We model budget selection as a subgoal-conditioned policy over discrete allocations. Specifically, at HL step $k$ we parameterize $b_\psi(\cdot \mid h_k, z_k) \in \Delta(\{0, 1, \ldots, B_k\})$, where $B_k$ denotes the remaining budget at step $k$, and the distribution is conditioned on the current latent state and the selected subgoal. This encourages context-aware resource allocation that can reflect each subgoal's temporal scale.

**Budget head as a learned policy (actor–critic).** We model budget selection as a stochastic policy over discrete allocations, conditioned on the current HL latent state and the *selected subgoal*. While we use a standard policy-gradient actor–critic objective similar in form to Feng et al. (2025), our key difference is *task-conditioned budgeting*: the agent first commits to *what* to do via $z_k$, and then allocates *how*

*much resource* to spend via $b_k \sim b_\psi(\cdot \mid h_k, z_k)$. This coupling lets budgets reflect the planned abstract task and its context, rather than being predicted as a subgoal-agnostic scalar.

**Search-backed return for budgeting.** We train the budget policy with a transition-conditioned target $\hat{G}_k^{(b)} = R_k + \gamma\, v^{\mathrm{TS}}(s_{k+1})$, where $v^{\mathrm{TS}}(s_{k+1})$ is the next-step search-backed root value from MCTS. See Appendix G for full details.

**High-level training objective.** We train the high-level planner with MuZero-style search-backed policy/value/reward losses, augmented with the MTS–SMDP geometric constraints and a budget actor–critic objective. Full definitions are provided in Appendix G.1.

### 4.4. Low-Level Policy for Combinatorial Actions

Once the high-level planner selects a subgoal $z_k$ and allocates a corresponding budget $b_k$, the low-level policy translates this directive into a sequence of primitive combinatorial actions. At this level, the agent must select from a vast, discrete action space under local information, and a purely myopic policy would fail to align its actions with the long-term plan.

Conditioning on $z_k$ is therefore important. The subgoal embedding communicates strategic intent and the implicit temporal scale learned by the multi-timescale world model, guiding the low-level policy on *what* to achieve. We implement the low-level policy as a subgoal-conditioned action-value network trained with a DQN objective. A GNN processes the current graph state, concatenating $z_k$ to each node's representation to compute Q-values for each primitive action (for example, node selection) in the context of the subgoal.

The network minimizes the Bellman error over transitions in a low-level replay buffer $\mathcal{D}_{LL}$:

$$\mathcal{L}_{LL} = \mathbb{E}_{(s_t', z, a_t, r_t, s_{t+1}') \sim \mathcal{D}_{LL}} \left[ \left( Q(s_t', a_t \mid z) - y_t \right)^2 \right],$$

with targets computed using a target network $Q'$ and the *primitive-time* discount $\gamma_{\mathrm{LL}}$: $y_t = r_t + \gamma_{\mathrm{LL}} \max_{a'} Q'(s_{t+1}', a' \mid z)$. During execution, the low-level policy uses this action-value function to generate primitive actions that pursue $z_k$, acting for up to $b_k$ steps and translating the high-level plan into a concrete combinatorial solution.

## 5. Experiments

### 5.1. Experimental Setup

**Domains** We evaluate LMTA on two canonical SSCO domains, using larger and more challenging problem instances

*Table 1.* AIM results on $N = 500$ graphs for SSCO-specific baselines and domain heuristics. Bold denotes the best mean. LMTA improves over WS-option with $p \le 0.05$.

| Method | $T = 10$ $K = 50$ | $T = 10$ $K = 60$ | $T = 10$ $K = 70$ | $T = 20$ $K = 10$ |
|---|---|---|---|---|
| **LMTA** | **286.11±4.05** | **303.50±2.52** | **324.15±2.38** | **161.71±1.39** |
| WS-option | 268.23±3.75 | 280.44±5.46 | 301.53±9.10 | 123.02±2.88 |
| Flat DQN | 255.56±4.15 | 171.80±3.16 | 243.46±3.78 | 111.46±2.95 |
| average-degree | 268.16±4.29 | 292.83±4.68 | 311.35±3.91 | 137.26±2.71 |
| average-score | 272.26±4.38 | 293.81±4.75 | 313.74±3.02 | 150.36±3.01 |
| normal-degree | 127.27±2.43 | 147.09±2.81 | 165.82±2.15 | 31.41±1.05 |
| normal-score | 134.31±2.59 | 155.74±2.99 | 172.95±2.33 | 34.43±1.12 |
| static-degree | 266.51±4.27 | 287.66±4.61 | 296.18±3.95 | 117.26±2.33 |
| static-score | 269.21±4.31 | 291.01±4.69 | 310.82±4.00 | 130.09±2.65 |

*Table 2.* SOP results on $N = 500$ graphs for SSCO-specific baselines and domain heuristics. Bold denotes the best mean. LMTA improves over WS-option with $p \le 0.05$.

| Method | $T = 10$ $K = 50$ | $T = 10$ $K = 60$ | $T = 10$ $K = 70$ | $T = 20$ $K = 10$ |
|---|---|---|---|---|
| **LMTA** | **29.79±0.46** | **30.39±0.67** | **31.60±1.94** | **13.51±0.45** |
| WS-option | 16.08±1.32 | 17.03±0.99 | 12.99±1.25 | 8.04±2.41 |
| Flat DQN | 18.44±0.18 | 18.99±1.06 | 15.90±0.67 | 7.62±0.22 |
| GA | 12.35±0.21 | 14.51±0.27 | 15.23±0.22 | 7.91±0.18 |
| Greedy | 17.17±0.22 | 21.91±0.25 | 23.58±0.16 | 4.89±0.15 |

than those used in prior work. To ensure a direct and fair comparison, the environments and problem parameters are scaled up but remain consistent in principle with Feng et al. (2025)[3].

- **Adaptive Influence Maximization (AIM):** A stochastic propagation problem on graphs, for which we use instances with $N = 500$ nodes.
- **Stochastic Orienteering Problem (SOP):** A stochastic route planning problem, for which we test on instances with $N = 500$ nodes.

**Additional domain.** We also evaluate on the **Power-2500** benchmark, which stresses larger-scale combinatorial structure; due to space, we report its full setup and results in Appendix N.

**Evaluation Protocol** To ensure a robust assessment, we evaluate all methods across a matrix of problem settings, varying both the episode horizon ($T$) and the total resource budget ($K$). Specifically, we test on configurations of $(T, K) \in \{(10, 50), (10, 60), (10, 70), (20, 10)\}$.

**Statistical reporting.** We run 10 random seeds per configuration and report **mean ± standard error of the mean (s.e.m.)**. $p$-values are computed from two-sided Welch $t$-tests on seed means.

**Implementation details and hyperparameters.** Unless otherwise stated, we use a shared set of training hyperparameters across the domains, summarized in Table 4 (Ap-

---

[3]Full implementation details for both environments are provided in Appendix D.

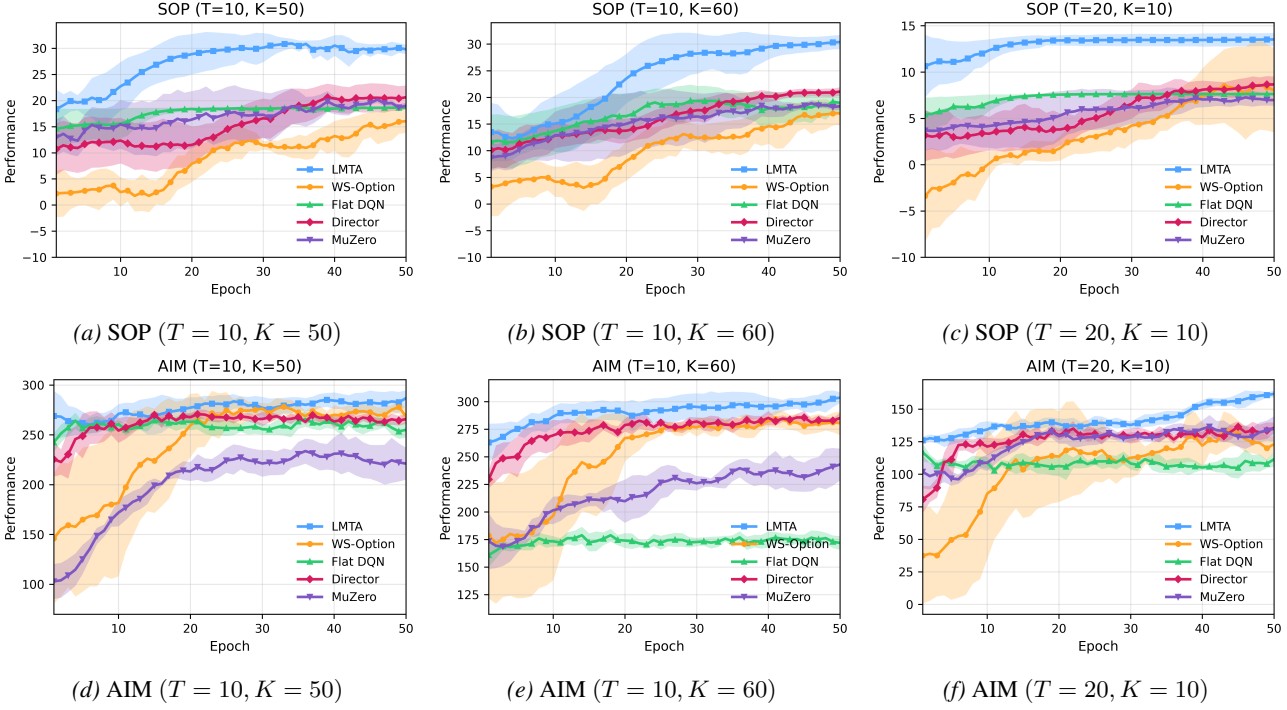

*Figure 2.* Training curves comparing LMTA against WS-Option, Flat DQN, Director, and Flat MuZero across AIM and SOP settings.

pendix F) and Table 5 (Appendix F). Hyperparameter sensitivity sweeps are provided in Appendix I.

**Baselines** We compare LMTA against: (i) the SSCO-specific WS-option (Feng et al., 2025), (ii) fixed-stride hierarchical model-based RL (Director (Hafner et al., 2022)), (iii) a flat latent-space planning baseline (MuZero (Schrittwieser et al., 2020)), and (iv) a flat model-free baseline (Flat DQN), alongside strong domain heuristics for AIM and SOP. Full baseline definitions and implementation details are provided in Appendix E.

### 5.2. Comparison with Baselines

As shown in Tables 1 and 2, LMTA outperforms the SSCO-specific baselines and domain heuristics across the tested configurations. We observed the performance gap widen as the problem complexity (for example, budget $K$) increases, highlighting the benefit of planning over variable-duration abstractions supported by the multi-timescale world model.

Figure 2 complements the tabular comparison by including Director and Flat MuZero, and further illustrates the advantages of our approach: LMTA not only converges to a higher final performance but also shows greater sample efficiency and stability throughout training, whereas WS-option often exhibits larger variance and slower convergence, especially in the SOP domain, which is consistent with the difficulty of credit assignment for model-free high-level policies in SMDPs. The Flat DQN baseline sometimes learns quickly

yet consistently plateaus at a suboptimal level, highlighting the importance of hierarchical abstraction for effective long-horizon planning in these complex settings. Although WS-option is a strong baseline in AIM, its performance degrades markedly on SOP, exposing a limitation of budget-only HRL in tasks requiring spatial reasoning: SOP is essentially a routing problem where the value of an action depends on the global path, so a scalar budget conveys *how long* to travel but not *where* to go, leaving the low-level policy to explore myopically. In contrast, LMTA's subgoals can capture *target regions* or *structural patterns*, *e.g.*, community clusters in AIM (Fig. 3a) and spatially dense neighborhoods in SOP (Fig. 3b, dashed circle), providing the spatial context that, in our analysis, helps the MCTS planner assemble globally coherent routes.

Beyond the main comparisons, Appendix N evaluates AIM at the $N = 200$ benchmark scale used by Feng et al. (2025) and reports a same-protocol scaling curve over $N \in \{200, 300, 400, 500\}$. These results show that LMTA's advantage is not specific to the larger $N = 500$ instances and remains consistent as graph size increases. Appendix N also reports complementary checks on Power-2500, tiny AIM instances where exhaustive enumeration of the stochastic decision tree is tractable, and Sampled MuZero (Hubert et al., 2021); these results respectively support LMTA's robustness on a larger graph, its closeness to the exact optimum when exact solution is tractable, and the need for variable-duration abstraction beyond primitive-action sam-

pling. Finally, Appendix O reports matched compute sweeps showing that LMTA remains strong even with little or no online search.

## 5.3. Assessing the Learned Policies

To rigorously dissect the contributions of LMTA's core architectural components, we conducted a comprehensive ablation study. We progressively build up from an option-based baseline (A) to the full LMTA model (E), quantifying the performance gain at each step. The results, presented in Table 3 for both the AIM ($N = 500, K = 70$) and SOP ($N = 500, K = 70$) domains, support our design choices and highlight the synergy between the framework's components.

*(a)* AIM Ablation ($N = 500, K = 70$).

| Algorithm Variant | Avg. Reward |
|---|---|
| A: Baseline | 301.53 ±9.10 |
| B: + LL Subgoal-Cond. | 309.52 ±3.71 |
| C: + Budget Subgoal-Cond. | 314.65 ±3.80 |
| D: + MTS (Fixed Margin) | 322.04 ±2.91 |
| **E: LMTA (Full Model)** | **324.15** ±**2.38** |
| F: LMTA-MF (Model-Free) | 307.29 ±5.62 |

*(b)* SOP Ablation ($N = 500, K = 70$).

| Algorithm Variant | Avg. Reward |
|---|---|
| A: Baseline | 12.99 ±1.25 |
| B: + LL Subgoal-Cond. | 17.03 ±1.98 |
| C: + Budget Subgoal-Cond. | 22.75 ±1.05 |
| D: + MTS (Fixed Margin) | 28.62 ±1.19 |
| **E: LMTA (Full Model)** | **31.60** ±**1.94** |
| F: LMTA-MF (Model-Free) | 15.89 ±1.91 |

*Table 3.* Ablation study results.

**From Budget-Only to Goal-Conditioned Execution** Our starting point is a strong option HRL baseline akin to *WS-option* (A), which uses a model-free high-level policy to allocate a budget but does not communicate a strategic subgoal. In our next variant (B), we introduce a MuZero planner that generates subgoals and conditions the low-level policy on them, while keeping the budget allocation independent of the subgoal. The performance gain from A to B demonstrates the immediate value of providing the low-level policy with strategic direction (*what* to do), even when the resource allocation (*how long* to act) is not yet aligned with that strategy. Note that at this stage, the MuZero planner is trained without the MTS objective.

**Aligning Resources with Strategy** In variant C, we take the logical next step by also conditioning the budget allocation on the selected subgoal. The performance improvement from B to C shows the benefit of this alignment. When the agent can allocate more resources to more ambitious subgoals and fewer to simple ones, its planning becomes more

efficient and effective. At this stage, we have a complete goal-conditioned HRL agent, but its world model remains a standard predictive model.

**The Critical Role of the MTS World Model** A clear performance improvement occurs when moving from variant C to D, where we introduce the *MTS objective with a fixed margin*. For this, we use the average budget times of the low level fixed margin $\kappa$. This variant uses our proposed push-pull dynamic to structure the latent space, but with a single, fixed temporal scale for all subgoals. This improvement underscores our central claim: a generic application of model-based planning may be insufficient for SMDPs. Explicitly structuring the latent space for multi-timescale reasoning is the critical component that unlocks effective planning over temporally abstract actions.

Moving from variant D to the full LMTA model (E), we replace the fixed margin with the proposed dynamic, learnable margin predictor. The resulting performance gain highlights the value of adaptive learning strategies. Allowing the agent to discover that different subgoals have different intrinsic temporal scales provides the final layer of strategic flexibility required to address these complex, multi-stage optimization problems.

**The Benefit of High-Level Planning** Finally, to isolate the contribution of model-based planning itself, we evaluate *LMTA-MF (Model-Free)* (F). This variant retains the full hierarchical and goal-conditioned structure of LMTA but replaces the MuZero planner with a reactive, model-free DQN policy. Its performance is clearly inferior to the full, planning-based model (E). This result confirms that in stochastic combinatorial domains, the foresight provided by the model-based planner is a key driver of performance, enabling the agent to find more robust and effective long-term strategies.

## 6. Conclusion

We introduce LMTA, a model-based hierarchical framework for sequential stochastic combinatorial decision-making. LMTA learns a multi-timescale SMDP world model that makes latent macro-transition magnitudes reflect subgoal-specific temporal scales, enabling tree-search planning over variable-duration abstractions without explicit duration prediction. Combined with a subgoal-conditioned budget policy for resource allocation, LMTA outperforms strong baselines on challenging SSCO benchmarks. These results suggest that SMDP-aware multi-timescale world models are a promising foundation for model-based HRL in other long-horizon, resource-constrained planning problems.

## Acknowledgements

We acknowledge CSC – IT Center for Science, Finland, for awarding this project access to the LUMI supercomputer, owned by the EuroHPC Joint Undertaking, hosted by CSC (Finland) and the LUMI consortium through CSC. We acknowledge the computational resources provided by the Aalto Science-IT project. We acknowledge funding from Research Council of Finland (353198, 357301).

## Impact Statement

This work develops general-purpose methodology for planning and learning in sequential stochastic combinatorial optimization, with experiments including Adaptive Influence Maximization (AIM) as a benchmark abstraction of diffusion processes on networks. Such methods can have beneficial applications, for example improving the allocation of limited resources in public health messaging, emergency response, or information campaigns aimed at increasing awareness of verified guidance. However, AIM-style optimization can also be misused to amplify misinformation, enable targeted persuasion, or exacerbate representational harms when deployed on human social networks. Our experiments use synthetic or benchmark instances and do not constitute deployment guidance; nevertheless, we acknowledge the dual-use nature of influence maximization and encourage future work to incorporate safeguards such as policy constraints, auditing and transparency mechanisms, and domain-specific oversight when applying these techniques in real-world sociotechnical settings.

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

# Appendix Contents

# A. Theoretical Analysis for the MTS-SMDP World Model

This appendix provides complete assumptions and detailed theoretical analysis. For completeness, we restate the full unified objective in Appendix A.1. All latent outputs of $h_\theta$ and $g_\theta$ are $\ell_2$-normalized to lie on the unit sphere $\mathbb{S}^{d-1}$, and $d(\cdot, \cdot)$ denotes the chordal distance on $\mathbb{S}^{d-1}$, as defined in the main text. We use $\mathbf{1}\{\cdot\}$ for indicator functions and $\{\mathcal{F}_i\}_{i \geq 0}$ for the filtration up to step $i$.

## A.1. Notation and Unified Loss

Let $h_\theta : \mathcal{S} \to \mathbb{R}^d$ and $g_\theta^h : \mathbb{R}^d \times \mathcal{Z} \to \mathbb{R}^d$ be the shared encoder and HL transition head; both outputs are $\ell_2$-normalized on forward pass to lie on $\mathbb{S}^{d-1}$. We write the chordal distance on the unit sphere as $d(\cdot, \cdot)$ (defined in the main text). Define the macro-step latent displacement

$$d_\theta(s_t', z) \;=\; d\big(h_\theta(s_t'),\, g_\theta^h(h_\theta(s_t'), z)\big), \qquad \sigma_{\theta, \phi}(s_t', z) \;=\; d_\theta(s_t', z) \;+\; m_\phi(z),$$

where $m_\phi$ is the learnable margin head (with an $\ell_2$ regularizer).

Let $\kappa > 0$ denote the per-step cap used in the Pull term; since $d(\cdot, \cdot)$ is chordal distance on $\mathbb{S}^{d-1}$, we have $d(u, v) \in [0, 2]$ and thus $\kappa \in (0, 2]$ in principle.

Let $(z, b)$ denote the chosen HL subgoal and budget. Executing $(z, b)$ from a high-level decision point corresponding to some primitive time $t$ induces a primitive rollout segment

$$(s_t', s_{t+1}', \ldots, s_{t+\tau}'),$$

with realized duration $\tau = \tau(s_t', z, b) \in \{0, \ldots, b\}$ (termination or budget exhaustion).

For two executed HL transitions $(s_i, z_i, b_i)$ and $(s_j, z_j, b_j)$ drawn from the decision-time experience distribution, let $\tau_i := \tau(s_i, z_i, b_i)$ and $\tau_j := \tau(s_j, z_j, b_j)$ denote their realized durations. We store pairwise labels

$$Y = \text{sign}\big(\tau_i - \tau_j\big) \in \{-1, 0, 1\}.$$

We write $\mu(s, z) := \mathbb{E}_{b \sim \pi_b(\cdot | s, z)}[\mathbb{E}[\tau(s, z, b) \mid s, z, b]]$ for the effective (budget-marginalized) mean duration, and use $\bar{\mu}(s, z) \equiv \mu(s, z)$ interchangeably when convenient.

**Budget policy notation.** We use $\pi_b(\cdot \mid s_t', z)$ to denote the learned budget policy. Throughout Appendix A, $\pi_b(\cdot \mid s_t', z) \equiv b_\psi(\cdot \mid h_\theta(s_t'), z)$, i.e., the same policy implemented as a function of the decision-time latent and the chosen subgoal. We use $\pi_b$ consistently to avoid notation drift.

We train with the unified loss:

$$\begin{aligned}
\mathcal{L}_{\text{unified}}(\theta, \phi) = \; & w_{\text{cap}} \, \mathbb{E}_{(s_t', s_{t+1}') \sim \mathcal{D}_{\text{LL}}} \left[ \big( d\big(h_\theta(s_t'), h_\theta(s_{t+1}')\big) - \kappa \big)_+^2 \right] \\
& + \; w_{\text{push}} \, \mathbb{E}_{(s_t', z)} \big[ \big(m_\phi(z) - d_\theta(s_t', z)\big)_+ \big] \\
& + \; w_{\text{order}} \, \mathbb{E}_{(s_i, z_i, b_i;\, s_j, z_j, b_j)} \Big[ \big(1 - Y \, \Delta\sigma\big)_+ \Big] \; + \; \lambda_m \|m_\phi\|_2^2,
\end{aligned} \tag{3}$$

where $(x)_+ = \max\{0, x\}$ and

$$\Delta\sigma := \sigma_{\theta, \phi}(s_i, z_i) - \sigma_{\theta, \phi}(s_j, z_j).$$

Unless otherwise specified, experiments use $w_{\text{cap}} = w_{\text{push}} = w_{\text{order}} = 0.10$.

**Executed-duration targets.** The duration signal $\tau(s, z, b)$ is the stopping time induced by the selected subgoal, allocated budget, low-level policy, and environment legality constraints. It is therefore the duration of the realized SMDP transition rather than an unconstrained completion time. This makes the temporal supervision consistent with the transitions used for planning. Empirically, the MTS loss ablations and Kendall-$\tau$ validation indicate that these executed-duration targets provide a useful ordering signal for the learned latent geometry.

## A.2. Assumptions

We distinguish the primitive-time discount $\gamma_{\mathrm{LL}}$ (used *inside* macro rewards $R_k$) from the decision-time discount $\gamma$ (used *between* high-level decisions in planning backups). The geometric results below do not depend on discounting; discounting matters only in the planning regret bound.

**Assumption A.1** (Budget-Marginalized Modeling Error on Planning Rollouts). Let $\mathcal{U}_{\mathrm{plan}} := \mathcal{U}_{\mathrm{dt}} \cup \mathcal{U}_{\mathrm{roll}}^{(H)}$. Define

$$\varepsilon_{\mathrm{dyn}} := \sup_{(u,z)\,:\,u\in\mathcal{U}_{\mathrm{plan}},\,z\in\mathcal{Z}} \mathbb{E}\big[d\big(g_\theta^h(u,z),\,c(u,z)\big)\big],$$

where $c(u,z)$ denotes the (encoded) next decision-time latent under executing $z$ from a decision-time state whose encoding equals $u$, under the endogenous budget policy and LL controller. We assume $\varepsilon_{\mathrm{dyn}} < \infty$ and set $\bar{\varepsilon} := \varepsilon_{\mathrm{dyn}}$.

*Remark* A.2 (Connecting $\varepsilon_{\mathrm{dyn}}$ to a regression objective). If $g_\theta$ is trained with a squared chordal regression objective

$$\mathcal{L}_{\mathrm{dyn}} = \mathbb{E}\Big[d\big(g_\theta^h(h_\theta(s),z),\,h_\theta(s^+)\big)^2\Big] \quad \text{(under the same budget-marginalized sampling of } s^+),$$

then Jensen implies $\mathbb{E}[d(\cdot)] \leq \sqrt{\mathbb{E}[d(\cdot)^2]}$, yielding

$$\varepsilon_{\mathrm{dyn}} \leq \sup_{(s,z)} \sqrt{\mathbb{E}\Big[d\big(g_\theta^h(h_\theta(s),z),\,h_\theta(s^+)\big)^2 \,\Big|\, s,z\Big]}.$$

Thus the modeling-error term appearing in later bounds is an explicit approximation quantity.

**Assumption A.3** (Cap-risk control (linked to $\mathcal{L}_{\mathrm{cap}}$)). Let $d_i := d\big(h_\theta(s'_{t+i}),\,h_\theta(s'_{t+i+1})\big)$ be the per-step chordal latent distance along any low-level rollout segment, and define the exceedance $X_i := (d_i - \kappa)_+$. Assume there exists a finite constant $\varepsilon_{\mathrm{cap}} \geq 0$ such that for all $i$,

$$\mathbb{E}\big[X_i^2 \mid \mathcal{F}_i\big] \leq \varepsilon_{\mathrm{cap}}^2 \qquad \text{a.s.,} \tag{4}$$

where $\{\mathcal{F}_i\}$ is the filtration up to step $i$.

**Connection to training.** If $(s'_t, s'_{t+1})$ are sampled from the same distribution as the on-policy primitive transitions, then $\varepsilon_{\mathrm{cap}}^2$ can be taken as (or upper bounded by) the population cap risk $\mathbb{E}\big[(d(h_\theta(s'_t), h_\theta(s'_{t+1})) - \kappa)_+^2\big]$, which is exactly $\mathcal{L}_{\mathrm{cap}}(\theta)$ in Equation (2a).

**Assumption A.4** (Informative Pairwise Labels (with ties)). Let $\mu(s,z) := \mathbb{E}_{b\sim\pi_b(\cdot|s,z)}\big[\mathbb{E}[\tau(s,z,b) \mid s,z,b]\big]$ denote the effective mean duration. Consider two executed HL transitions $(s_i, z_i, b_i)$ and $(s_j, z_j, b_j)$ with realized durations $\tau_i = \tau(s_i, z_i, b_i)$ and $\tau_j = \tau(s_j, z_j, b_j)$, and define $Y = \mathrm{sign}(\tau_i - \tau_j) \in \{-1, 0, 1\}$. We call the pair *informative* if $\Pr(Y \neq 0 \mid s_i, z_i, s_j, z_j) > 0$ and $\mu(s_i, z_i) \neq \mu(s_j, z_j)$. For any informative pair,

$$\Pr\{Y = 1 \mid Y \neq 0,\, s_i, z_i, s_j, z_j\} > \tfrac{1}{2} \quad \Longleftrightarrow \quad \mu(s_i, z_i) > \mu(s_j, z_j).$$

*Remark* A.5 (Label-noise model assumption). Assumption A.4 is a monotonic label-noise condition: sampled duration comparisons are directionally consistent (in probability) with the ordering of effective mean durations. This is required to guarantee that minimizing the noisy pairwise hinge recovers the correct ranking on informative pairs.

**Assumption A.6** (Standard Regularity and Value Smoothness). (i) Primitive rewards are bounded: $r_t \in [0, r_{\max}]$, with primitive discount $\gamma_{\mathrm{LL}} \in [0, 1)$.

(ii) **Local value smoothness in the learned latent geometry (Lipschitz; $\alpha = 1$).** Let $h_\theta$ encode decision-time primitive states onto $\mathbb{S}^{d-1}$, and define the set of latents that can appear in depth-$H$ planning as

$$\mathcal{U}_{\mathrm{dt}} := \{\, h_\theta(s) \,:\, s \text{ is a decision-time primitive state} \,\},$$

and the depth-$H$ modeled rollout set

$$\mathcal{U}_{\mathrm{roll}}^{(H)} := \Big\{\, (g_\theta^h)^{(k)}\big(h_\theta(s), z_{0:k-1}\big) \,:\, s \text{ decision-time},\, k \in \{1, \ldots, H\},\, z_i \in \mathcal{Z} \,\Big\},$$

where $(g_\theta^h)^{(1)}(u, z_0) := g_\theta^h(u, z_0)$ and $(g_\theta^h)^{(k+1)}(u, z_{0:k}) := g_\theta^h((g_\theta^h)^{(k)}(u, z_{0:k-1}), z_k)$.

We assume there exists a function $V_{\text{lat}} : \mathbb{S}^{d-1} \to \mathbb{R}$ such that the true optimal decision-time value satisfies

$$V(s) = V_{\text{lat}}(h_\theta(s)) \qquad \text{for all decision-time states } s,$$

and $V_{\text{lat}}$ is $L_S$-Lipschitz w.r.t. the chordal metric on the latent region relevant to planning: for all $u, v \in \mathcal{U}_{\text{dt}} \cup \mathcal{U}_{\text{roll}}^{(H)}$,

$$\left| V_{\text{lat}}(u) - V_{\text{lat}}(v) \right| \leq L_S \, d(u, v).$$

*Remark (scope and interpretation).* This is a regularity condition on the *environment value composed with the representation* (i.e., $V$ factors through $h_\theta$ at decision times), not a Lipschitz claim about any parametric value head. It only needs to hold on $\mathcal{U}_{\text{dt}} \cup \mathcal{U}_{\text{roll}}^{(H)}$ (the latents that can appear as true decision-time latents or depth-$H$ modeled rollouts during MCTS), not on the entire sphere.

*Remark (Hölder extension, optional).* If instead $V_{\text{lat}}$ is Hölder with exponent $\alpha \in (0, 1]$ on $\mathcal{U}_{\text{dt}} \cup \mathcal{U}_{\text{roll}}^{(H)}$, i.e. $|V_{\text{lat}}(u) - V_{\text{lat}}(v)| \leq L_S \, d(u, v)^\alpha$, then the model-bias term $\gamma L_S \bar{\varepsilon}$ in Theorem A.13 tightens to $\gamma L_S \bar{\varepsilon}^\alpha$.

**Assumption A.7** (Macro-reward prediction error). Let $R(s, z)$ denote the realized macro reward when executing $(z, b)$ with $b \sim \pi_b(\cdot \mid s, z)$ and the learned low-level controller. Assume the macro reward head $R_\theta(s, z)$ satisfies

$$\varepsilon_R := \sup_{(s, z)} \left| \mathbb{E}[R(s, z) \mid s, z] - R_\theta(s, z) \right| < \infty.$$

**Proposition A.8** (Decision-time discount approximation). *LMTA uses a fixed high-level discount $\gamma$ rather than the primitive-time continuation factor $\gamma_{\text{LL}}^\tau$. Define*

$$Q_{\text{prim}}(s, z) := \mathbb{E}\left[ R_\tau(s, z) + \gamma_{\text{LL}}^\tau V(s^+) \mid s, z \right], \qquad Q_{\text{dt}}(s, z) := \mathbb{E}\left[ R_\tau(s, z) + \gamma V(s^+) \mid s, z \right],$$

*where $R_\tau(s, z)$ includes within-macro primitive-time discounting. If $|V(s)| \leq V_{\max}$ and $\gamma = \gamma_{\text{LL}}^{\bar{\mu}}$ for a reference duration $\bar{\mu}$, then*

$$|Q_{\text{prim}}(s, z) - Q_{\text{dt}}(s, z)| \leq V_{\max} |\log \gamma_{\text{LL}}| \sqrt{\text{Var}(\tau \mid s, z) + (\mu(s, z) - \bar{\mu})^2}.$$

*Thus the fixed decision-time backup is closest to the primitive-time SMDP backup when durations are concentrated around the reference duration $\bar{\mu}$.*

*Proof.* By definition,

$$|Q_{\text{prim}}(s, z) - Q_{\text{dt}}(s, z)| \leq V_{\max} \, \mathbb{E}[|\gamma_{\text{LL}}^\tau - \gamma| \mid s, z].$$

Choosing $\gamma = \gamma_{\text{LL}}^{\bar{\mu}}$ and using that $x \mapsto \gamma_{\text{LL}}^x$ is $|\log \gamma_{\text{LL}}|$-Lipschitz gives

$$|\gamma_{\text{LL}}^\tau - \gamma| = \left| \gamma_{\text{LL}}^\tau - \gamma_{\text{LL}}^{\bar{\mu}} \right| \leq |\log \gamma_{\text{LL}}| \, |\tau - \bar{\mu}|.$$

Therefore,

$$|Q_{\text{prim}}(s, z) - Q_{\text{dt}}(s, z)| \leq V_{\max} |\log \gamma_{\text{LL}}| \, \mathbb{E}[|\tau - \bar{\mu}| \mid s, z].$$

By Cauchy–Schwarz,

$$\mathbb{E}[|\tau - \bar{\mu}| \mid s, z] \leq \sqrt{\text{Var}(\tau \mid s, z) + (\mu(s, z) - \bar{\mu})^2},$$

which proves the claim. $\square$

### A.3. Deriving Macro–Micro Control from the Objective

We first show that minimizing the Pull term imposes an upper bound on decision-time latent displacement proportional to execution duration. Unlike a trivial bound that follows only from bounded distances on $\mathbb{S}^{d-1}$, our bound explicitly improves as the cap-risk $\mathcal{L}_{\text{cap}}$ decreases (Assumption A.3).

**Lemma A.9** (Expectation-Level Macro–Micro Control (cap-risk form)). *Under Assumptions A.1 and A.3, for any decision-time state $s_t'$ and subgoal $z$, with $b \sim \pi_b(\cdot \mid s_t', z)$ and realized duration $\tau = \tau(s_t', z, b)$,*

$$\mathbb{E}\left[ d\big(h_\theta(s_t'), \, h_\theta(s_{t+\tau}')\big) \mid s_t', z \right] \leq \mathbb{E}[\tau \mid s_t', z] \, (\kappa + \varepsilon_{\text{cap}}), \tag{5}$$

*and the budget-marginalized model–chord gap satisfies*

$$\left| \mathbb{E}[d_\theta(s_t', z) \mid s_t', z] - \mathbb{E}\big[d\big(h_\theta(s_t'), h_\theta(s_{t+\tau}')\big) \mid s_t', z\big] \right| \leq \bar{\varepsilon}. \tag{6}$$

*Proof.* Fix $s_t', z$. Let $\tau$ be the realized duration under $b \sim \pi_b(\cdot \mid s_t', z)$. By the triangle inequality for the metric $d$,

$$d\big(h_\theta(s_t'), h_\theta(s_{t+\tau}')\big) \ \leq \ \sum_{i=0}^{\tau-1} d\big(h_\theta(s_{t+i}'), h_\theta(s_{t+i+1}')\big).$$

Define $d_i := d(h_\theta(s_{t+i}'), h_\theta(s_{t+i+1}'))$ and $X_i := (d_i - \kappa)_+$, so that $d_i \leq \kappa + X_i$. Hence

$$d\big(h_\theta(s_t'), h_\theta(s_{t+\tau}')\big) \ \leq \ \sum_{i=0}^{\tau-1}(\kappa + X_i) = \tau\kappa + \sum_{i=0}^{\tau-1} X_i.$$

Taking conditional expectation and using the tower property,

$$\mathbb{E}\left[\sum_{i=0}^{\tau-1} X_i \mid s_t', z\right] = \mathbb{E}\left[\sum_{i=0}^{\infty} \mathbf{1}\{i < \tau\}\, \mathbb{E}[X_i \mid \mathcal{F}_i] \ \middle|\ s_t', z\right].$$

By conditional Jensen and Assumption A.3,

$$\mathbb{E}[X_i \mid \mathcal{F}_i] \leq \sqrt{\mathbb{E}[X_i^2 \mid \mathcal{F}_i]} \leq \varepsilon_{\mathrm{cap}} \qquad \text{a.s.}$$

Therefore

$$\mathbb{E}\left[\sum_{i=0}^{\tau-1} X_i \mid s_t', z\right] \leq \varepsilon_{\mathrm{cap}}\, \mathbb{E}[\tau \mid s_t', z],$$

which together with the earlier bound implies Equation (5).

For Equation (6), let $a = h_\theta(s_t')$, $b' = g_\theta^h(h_\theta(s_t'), z)$ and $c = h_\theta(s_{t+\tau}')$. By the metric inequality $|d(a, b') - d(a, c)| \leq d(b', c)$ and the definition of $\bar\varepsilon$ in Assumption A.1,

$$|\mathbb{E}[d(a, b') \mid s_t', z] - \mathbb{E}[d(a, c) \mid s_t', z]| \leq \mathbb{E}[d(b', c) \mid s_t', z] \leq \bar\varepsilon.$$

$\square$

### A.4. Duration Lower Bound from the Learned Margin

We now show that combining the Push term with Lemma A.9 establishes the learned margin as a lower bound for the effective expected duration, up to (i) the *push residual* and (ii) the decision-time modeling error $\bar\varepsilon$.

**Proposition A.10** (Effective Expected Duration Lower Bound (push-residual form))**.** *Fix a decision-time state $s_t'$ and subgoal $z$. Define the* push residual *at $(s_t', z)$ by*

$$\rho(s_t', z) \ := \ \big(m_\phi(z) - d_\theta(s_t', z)\big)_+.$$

*Under Assumptions A.1 and A.3, the effective mean duration $\bar\mu(s_t', z) := \mathbb{E}[\tau \mid s_t', z]$ satisfies*

$$\boxed{\ \bar\mu(s_t', z) \ \geq \ \frac{m_\phi(z) - \rho(s_t', z) - \bar\varepsilon}{\kappa + \varepsilon_{\mathrm{cap}}}\ } \tag{7}$$

*(where the bound is informative when $m_\phi(z) > \rho(s_t', z) + \bar\varepsilon$).*

*Proof.* By definition of $\rho(s_t', z)$, we have the pointwise inequality

$$m_\phi(z) \ \leq \ d_\theta(s_t', z) + \rho(s_t', z).$$

Taking conditional expectation given $(s_t', z)$ gives

$$m_\phi(z) - \rho(s_t', z) \ \leq \ \mathbb{E}[d_\theta(s_t', z) \mid s_t', z].$$

By Lemma A.9 (model–chord gap Equation (6)),

$$\mathbb{E}[d_\theta(s'_t, z) \mid s'_t, z] \;\leq\; \mathbb{E}\big[d\big(h_\theta(s'_t), h_\theta(s'_{t+\tau})\big) \mid s'_t, z\big] + \bar{\varepsilon}.$$

Combining yields

$$m_\phi(z) - \rho(s'_t, z) - \bar{\varepsilon} \;\leq\; \mathbb{E}\big[d\big(h_\theta(s'_t), h_\theta(s'_{t+\tau})\big) \mid s'_t, z\big].$$

Apply Lemma A.9 (macro–micro bound Equation (5)) to obtain

$$m_\phi(z) - \rho(s'_t, z) - \bar{\varepsilon} \;\leq\; \bar{\mu}(s'_t, z)\,(\kappa + \varepsilon_{\mathrm{cap}}),$$

and rearrange. $\qquad\square$

**Population corollary (explicit link to $\mathcal{L}_{\mathrm{push}}$).** Let $\nu$ be the decision-time state distribution. Then $\mathbb{E}_{s\sim\nu}[\rho(s, z)] = \mathbb{E}_{s\sim\nu}[(m_\phi(z) - d_\theta(s, z))_+]$ is precisely the per-subgoal push hinge residual. Consequently,

$$\mathbb{E}_{s\sim\nu}[\bar{\mu}(s, z)] \;\geq\; \frac{m_\phi(z) - \mathbb{E}_{s\sim\nu}[\rho(s, z)] - \bar{\varepsilon}}{\kappa + \varepsilon_{\mathrm{cap}}}.$$

### A.5. Order Consistency via Pairwise Hinge

Finally, we show that the **Budget-Aware Order Consistency** term refines the coupling by ensuring that a population-optimal scorer orders subgoals consistently with their effective mean durations on informative pairs.

**Theorem A.11** (Budget-Aware Order Consistency (unconstrained population optimum))**.** *Let $f$ range over all measurable scoring functions over decision-time pairs $(s, z)$. Let $f^\star$ be a population minimizer of the pairwise hinge risk*

$$\mathcal{R}_{\mathrm{pair}}(f) \;=\; \mathbb{E}\Big[\max\{0,\, 1 - Y\,(f(s_i, z_i) - f(s_j, z_j))\}\Big],$$

*where the expectation is over the joint distribution of $(s_i, z_i, b_i, s_j, z_j, b_j, Y)$ with $Y \in \{-1, 0, 1\}$. Under Assumption A.4, for almost every informative pair $(s_i, z_i;\, s_j, z_j)$,*

$$\mathrm{sign}\big(f^\star(s_i, z_i) - f^\star(s_j, z_j)\big) \;=\; \mathrm{sign}\big(\mu(s_i, z_i) - \mu(s_j, z_j)\big).$$

*Proof.* Fix an informative pair $(s_i, z_i;\, s_j, z_j)$. Let $p_+ := \Pr(Y = 1 \mid s_i, z_i, s_j, z_j)$, $p_- := \Pr(Y = -1 \mid s_i, z_i, s_j, z_j)$, and $p_0 := \Pr(Y = 0 \mid s_i, z_i, s_j, z_j)$, with $p_+ + p_- > 0$ by informativeness. The conditional risk as a function of the scalar margin $u := f(s_i, z_i) - f(s_j, z_j)$ is

$$\varphi(u) \;=\; p_+\,(1 - u)_+ \;+\; p_-\,(1 + u)_+ \;+\; p_0\,(1 - 0 \cdot u)_+.$$

Since $(1 - 0 \cdot u)_+ = 1$ is constant in $u$, the $p_0$ term does not affect the minimizer. Define

$$\eta \;:=\; \Pr(Y = 1 \mid Y \neq 0,\, s_i, z_i, s_j, z_j) \;=\; \frac{p_+}{p_+ + p_-} \in (0, 1).$$

Then minimizing $\varphi(u)$ is equivalent to minimizing

$$\tilde{\varphi}(u; \eta) \;=\; \eta\,(1 - u)_+ \;+\; (1 - \eta)\,(1 + u)_+.$$

The function $\tilde{\varphi}(\cdot; \eta)$ is convex and piecewise linear. Analyzing its slope on the regions $u \leq -1$, $-1 < u < 1$, and $u \geq 1$ shows that a minimizer satisfies

$$u^\star \in \arg\min_u \tilde{\varphi}(u; \eta) \quad \Rightarrow \quad \mathrm{sign}(u^\star) = \mathrm{sign}(2\eta - 1).$$

By Assumption A.4, $\eta > \frac{1}{2}$ iff $\mu(s_i, z_i) > \mu(s_j, z_j)$ and $\eta < \frac{1}{2}$ iff the reverse holds. Thus the sign of the population-optimal margin coincides with the sign of the effective mean-duration difference on informative pairs. $\qquad\square$

*Remark* A.12 (Parametric approximation for $\sigma_{\theta,\phi}$). In practice $f_{\theta,\phi}(s, z) = d_\theta(s, z) + m_\phi(z)$ lies in a restricted hypothesis class. Then the same conclusion holds up to an approximation term: if $\mathcal{R}_{\mathrm{pair}}(f_{\theta,\phi}) - \inf_f \mathcal{R}_{\mathrm{pair}}(f) \leq \epsilon_\mathcal{F}$, the measure of mis-ordered informative pairs is controlled by $\epsilon_\mathcal{F}$ via standard surrogate calibration arguments for hinge ranking.

## A.6. Planning Regret for Depth-$H$ MCTS

For readability in this subsection, we drop the explicit primitive index and write $s$ for a generic *decision-time* environment state (equivalently, $s = s'_t$ for some primitive time $t$), and write $s^+$ for the subsequent decision-time state reached after executing a macro-action (equivalently, $s^+ = s'_{t+\tau}$). We use $V(s, z)$ to denote the *macro-action value* (depth-$H$ return) of choosing subgoal $z$ at decision-time state $s$.

We write $N(s, z)$ for the number of root visits allocated to action $z$ by the finite-simulation search at state $s$, and define

$$N_{\min}(s) := \min_{z \in \mathcal{Z}} N(s, z).$$

**Theorem A.13** (Planning error bound (with reward + transition mismatch)). *Consider depth-$H$ tree search over subgoals using $N$ simulations per decision and a MuZero-style PUCT rule. Assume: (i) the leaf evaluator $V_b$ satisfies $\|V_b - V\|_\infty \leq \varepsilon_V$; (ii) Assumptions A.1, A.6(ii), and A.7 hold; (iii) the root-action estimator $Q_b(s, \cdot)$ obeys the UCT-style concentration property: for any $\delta \in (0, 1)$, with probability at least $1 - \delta$,*

$$\sup_{z \in \mathcal{Z}} \left| Q_b(s, z) - \mathbb{E}[Q_b(s, z)] \right| \leq c_0 V_{\max}^{(H)} \sqrt{\frac{\log(|\mathcal{Z}|/\delta)}{N_{\min}(s)}}, \tag{8}$$

*for an absolute constant $c_0 > 0$.*

*Let $B_{\max}$ be an a priori upper bound on macro duration, so that $\tau(s, z, b) \leq B_{\max}$ almost surely. With primitive rewards bounded by $r_t \in [0, r_{\max}]$ (Assumption A.6(i)), the macro reward satisfies*

$$R(s, z) = \sum_{i=0}^{\tau-1} \gamma_{\mathrm{LL}}^i r_{t+i} \leq \sum_{i=0}^{B_{\max}-1} \gamma_{\mathrm{LL}}^i r_{\max} \leq \frac{r_{\max}}{1 - \gamma_{\mathrm{LL}}} =: R_{\max}.$$

*Define $V_{\max}^{(H)} := \frac{1-\gamma^H}{1-\gamma} R_{\max}$.*

*Let $\hat{z} \in \arg\max_{z \in \mathcal{Z}} Q_b(s, z)$ be the (random) subgoal selected by finite-simulation search and $z^\star \in \arg\max_{z \in \mathcal{Z}} V(s, z)$ be optimal. Then, with probability at least $1 - \delta$ over the internal randomness of the finite-simulation tree search,*

$$V(s, z^\star) - V(s, \hat{z}) \leq 2\gamma^H \varepsilon_V + 2\frac{1-\gamma^H}{1-\gamma}\left(\varepsilon_R + \gamma L_S \bar{\varepsilon}\right) + 2c_0 V_{\max}^{(H)} \sqrt{\frac{\log(|\mathcal{Z}|/\delta)}{N_{\min}(s)}}. \tag{9}$$

*Proof.* We decompose the planning error by comparing the true depth-$H$ return, a modeled return, and the empirical MCTS estimate. Let $Q^\star(s, z)$ denote the true depth-$H$ return for selecting $z$ at the root and acting optimally thereafter in the true process. Let $Q^{\mathrm{mod}}(s, z)$ denote the depth-$H$ return under the modeled latent dynamics that advance deterministically to $g_\theta^h(h_\theta(s), z)$, with macro reward $R_\theta(s, z)$ and decision-time discount $\gamma$. Let $Q_b(s, z)$ be the empirical MCTS estimate using leaf evaluation $V_b$ at depth $H$. For any $z$,

$$Q^\star(s, z) - Q_b(s, z) = \underbrace{Q^\star(s, z) - Q^{\mathrm{mod}}(s, z)}_{\text{model bias}} + \underbrace{Q^{\mathrm{mod}}(s, z) - \mathbb{E}[Q_b(s, z)]}_{\text{leaf-evaluation bias}} + \underbrace{\mathbb{E}[Q_b(s, z)] - Q_b(s, z)}_{\text{Monte Carlo deviation}}.$$

*(i) Model bias.* We bound $Q^\star - Q^{\mathrm{mod}}$ using a one-step Bellman decomposition (no contractivity/non-expansiveness of $g_\theta$ is required). At any decision-time step, the reward mismatch is at most $\varepsilon_R$ by Assumption A.7. For the continuation value, let $c := h_\theta(s^+)$ be the true next decision-time latent and $u := g_\theta(h_\theta(s), z)$ be the modeled next latent. By Assumption A.6(ii),

$$\left| \mathbb{E}[V_{\mathrm{lat}}(c) \mid s, z] - V_{\mathrm{lat}}(u) \right| \leq \mathbb{E}\left[ |V_{\mathrm{lat}}(c) - V_{\mathrm{lat}}(u)| \mid s, z \right] \leq L_S \mathbb{E}\left[ d(c, u) \mid s, z \right].$$

By Assumption A.1 (with $\bar{\varepsilon} = \varepsilon_{\mathrm{dyn}}$), we have $\mathbb{E}[d(c, u) \mid s, z] \leq \bar{\varepsilon}$, hence

$$\left| \mathbb{E}[V_{\mathrm{lat}}(c) \mid s, z] - V_{\mathrm{lat}}(u) \right| \leq L_S \bar{\varepsilon}.$$

Therefore, the one-step Bellman backup mismatch is bounded by $\varepsilon_R + \gamma L_S \bar{\varepsilon}$. Summing discounted one-step errors over horizon $H$ yields

$$\sup_{z \in \mathcal{Z}} \left| Q^\star(s, z) - Q^{\mathrm{mod}}(s, z) \right| \leq \sum_{k=0}^{H-1} \gamma^k \left( \varepsilon_R + \gamma L_S \bar{\varepsilon} \right) = \frac{1-\gamma^H}{1-\gamma} \left( \varepsilon_R + \gamma L_S \bar{\varepsilon} \right).$$

*(ii) Leaf-evaluation bias (explicit horizon decay).* The leaf value is used at depth $H$ and is discounted by $\gamma^H$, hence

$$\left|Q^{\mathrm{mod}}(s,z) - \mathbb{E}[Q_b(s,z)]\right| \leq \gamma^H \|V_b - V\|_\infty \leq \gamma^H \varepsilon_V.$$

*(iii) Monte Carlo deviation.* Apply the concentration property (8).

Combining (i)–(iii), on the event in (8) we obtain

$$\sup_{z \in \mathcal{Z}} \left|Q^\star(s,z) - Q_b(s,z)\right| \leq \frac{1-\gamma^H}{1-\gamma}(\varepsilon_R + \gamma L_S \bar{\varepsilon}) + \gamma^H \varepsilon_V + c_0 V_{\max}^{(H)} \sqrt{\frac{\log(|\mathcal{Z}|/\delta)}{N_{\min}(s)}}.$$

Let $\hat{z} \in \arg\max_z Q_b(s,z)$ and $z^\star \in \arg\max_z Q^\star(s,z)$. Then

$$Q^\star(s,z^\star) - Q^\star(s,\hat{z}) \leq 2 \sup_{z \in \mathcal{Z}} \left|Q^\star(s,z) - Q_b(s,z)\right|.$$

Finally, $Q^\star(s,z) = V(s,z)$ by definition of the decision-time (budget-marginalized) SMDP value, so substituting the previous bound yields (9). $\qquad\square$

*Remark* A.14 (Recovering a bound in terms of total simulations $N$). If the realized search allocation satisfies a root-coverage condition

$$N_{\min}(s) \geq \alpha \frac{N}{|\mathcal{Z}|} \quad \text{for some } \alpha \in (0,1],$$

then the deviation term simplifies (up to constants) to

$$V_{\max}^{(H)} \sqrt{\frac{\log(|\mathcal{Z}|/\delta)}{N_{\min}(s)}} \leq V_{\max}^{(H)} \sqrt{\frac{|\mathcal{Z}| \log(|\mathcal{Z}|/\delta)}{\alpha N}}.$$

**Summary of Theoretical Results.**

1. **Lemma A.9** establishes expectation-level macro–micro control: the (budget-marginalized) expected decision-time latent displacement is upper-bounded by the (budget-marginalized) expected duration, scaled by $(\kappa + \varepsilon_{\mathrm{cap}})$, where $\varepsilon_{\mathrm{cap}}$ is controlled by the Pull (cap-risk) term via Assumption A.3.

2. **Proposition A.10** combines this with the *High-Level Margin (Push)* loss to show that the learned margin $m_\phi(z)$ yields a lower bound on the effective mean duration $\bar{\mu}(s'_t, z)$ up to approximation terms ($\bar{\varepsilon} = \varepsilon_{\mathrm{dyn}}$ from Assumption A.1 and slack $\rho$). This *derives* the geometry–duration correlation rather than assuming it.

3. **Theorem A.11** shows that the *Order Consistency* loss yields correct ranking of subgoals by effective duration on informative pairs, with explicit handling of tie labels $Y = 0$.

4. **Theorem A.13** provides a planning error bound that decomposes suboptimality into value approximation error, model bias, and a finite-simulation deviation term. The deviation term is stated via an explicit UCT-style concentration condition in terms of realized root visit counts $N_{\min}(s)$ (and can be related to total simulations $N$ under an additional root-coverage condition).

5. **Proposition A.8** bounds the difference between fixed decision-time and primitive-time SMDP backups in terms of duration variability and mismatch from the reference duration.

## B. Background on Latent-Space Tree Search Planning in LMTA

This section describes the latent-space tree-search planner used in LMTA, following the representation–dynamics–prediction paradigm of Schrittwieser et al. (2020). Unlike planning over primitive actions, our search branches over abstract subgoals $z \in \mathcal{Z}$.

Each simulation consists of three phases:

**1. Selection**   The search starts at the root latent $h^0 = h_\theta(s_{\text{env}})$ and traverses the tree by selecting the subgoal that maximizes a PUCT score:

$$z^k = \arg\max_{z \in \mathcal{Z}} \left[ Q(h^k, z) \; + \; c(h^k)\, P(h^k, z)\, \frac{\sqrt{\sum_{z'} N(h^k, z')}}{1 + N(h^k, z)} \right],$$

where $Q(h, z)$ is the current action-value estimate, $P(h, z)$ is the prior from the policy head, $N(h, z)$ is the visit count, and $c(h^k)$ is the usual exploration coefficient (implemented as in Schrittwieser et al. (2020)).

**2. Expansion and Recurrent Inference**   At an unexpanded leaf, we apply the dynamics model to simulate an SMDP transition in latent space:

$$h^{l+1}, \hat{R}^l = g_\theta(h^l, z^l),$$

and evaluate the new node using the prediction head:

$$\mathbf{p}^{l+1}, v^{l+1} = f_\theta(h^{l+1}).$$

**3. Backup**   We back up returns along the simulated trajectory using a *decision-time* discount $\gamma \in [0, 1)$. For a node at depth $k < l$, we use

$$G_d \;=\; \sum_{j=0}^{l-1-d} \gamma^j\, \hat{R}_{d+j} \;+\; \gamma^{l-d}\, v_{l+1}, \qquad d < l,$$

and update $Q(h^k, z^k)$ by a moving average with visit counts.

After $N_{\text{sim}}$ simulations, the search policy at the root is obtained from visit counts, e.g., $\pi_{\text{TS}}(z) \propto N(h^0, z)^{1/T_{\text{temp}}}$.

## C. Algorithm

## D. Environments

We evaluate on two SSCO benchmarks following Feng et al. (2025): Adaptive Influence Maximization (AIM) and Stochastic Orienteering (SOP). The environments define primitive states, stochastic transitions, rewards, and feasibility constraints. Our hierarchical agent interacts with them by selecting a high-level action $(z, b)$ at a decision time, where $z$ is a learned subgoal and $b$ is a budget allocation, and then executing primitive actions for up to $b$ steps before the next high-level decision.

**Adaptive Influence Maximization (AIM).**   AIM extends classical influence maximization to progressive, uncertain settings under the Independent Cascade (IC) model. The environment is a directed graph $G = (V, E)$. Each node is inactive/active/removed. When a node becomes active (seeded or activated), it attempts to activate each inactive neighbor on the next day with probability $p(\cdot)$, after which it cannot activate further. Following Feng et al. (2025), edge probabilities are set as a function of node indegree, and graphs are randomly generated. A total seed budget and a time horizon are given; before each day, the agent selects seed nodes subject to the remaining budget. The objective is to maximize the total number of influenced nodes within the horizon.

At a high-level decision time corresponding to day $t$, the agent commits to $(z_t, b_t)$, and the low level selects up to $b_t$ seed nodes sequentially conditioned on $z_t$ and the current graph state (with illegal-action masking for already active/removed nodes). The environment then applies one IC propagation step. The per-decision reward is the increase in influenced nodes induced during that day (including newly seeded and newly activated nodes), and the state is updated by the resulting node-status transitions and remaining budget.

**Stochastic Orienteering (SOP).**   SOP is a stochastic route-planning variant with a set of cities whose coordinates are sampled i.i.d., time-varying per-city profits, and a daily travel-distance limit with penalties for exceedance. The agent must choose which cities to visit over a fixed number of days, starting and ending at a designated city, to maximize total profit. Following Feng et al. (2025), a submodular aggregation is used for the one-day profit over selected cities; profits reset to zero upon visit and otherwise evolve stochastically.

At a high-level decision time for day $t$, the agent commits to $(z_t, b_t)$, and the low level selects a sequence of cities conditioned on $z_t$ for up to $b_t$ primitive selections. The transition updates the current location, applies the travel-distance rule (including

---

**Algorithm 1** LMTA Training Loop

---

1: Initialize networks: $h_\theta, f_\theta, g_\theta, m_\phi, b_\psi$, and LL policy $\pi^L$.
2: Initialize replay buffers $\mathcal{D}_{\text{HL}}$ (for episodes) and $\mathcal{D}_{\text{LL}}$ (for primitive transitions).
3: **for** each training epoch **do**
4:     **// Phase 1: Self-play experience collection**
5:     Run one full episode with horizon $T$ (high-level decisions).
6:     Let $\{t_k\}_{k=0}^{T-1}$ be the primitive times at which high-level decisions are taken (with $t_0 = 0$).
7:     **for** each high-level decision index $k = 0, 1, \ldots, T-1$ **do**
8:         Observe HL state $s_k$.
9:         Run tree search using $h_\theta, f_\theta, g_\theta$ to get search policy $\pi_k$ and value targets.
10:        Sample subgoal $z_k \sim \pi_k(\cdot)$.
11:        Sample budget $b_k \sim b_\psi(\cdot \mid h_\theta(s_k), z_k)$.
12:        Execute LL policy for up to $b_k$ primitive steps starting at time $t_k$; observe realized duration $\tau_k$ and next HL state $s_{k+1}$.
13:        Compute macro reward $R_k = \sum_{i=0}^{\tau_k - 1} \gamma_{\text{LL}}^i r_{t_k + i}$.
14:        Store HL transition $(s_k, z_k, b_k, R_k, s_{k+1})$ and search targets $(\pi_k, v_k)$ in the episode history.
15:        Store primitive transitions $(s'_t, z_k, a_t, r_t, s'_{t+1})$ for $t = t_k, \ldots, t_k + \tau_k - 1$ in $\mathcal{D}_{\text{LL}}$.
16:     **end for**
17:     Add the completed episode history to $\mathcal{D}_{\text{HL}}$.
18:     **// Phase 2: High-level updates**
19:     **if** $|\mathcal{D}_{\text{HL}}|$ and $|\mathcal{D}_{\text{LL}}|$ are large enough **then**
20:        Sample a batch of episode segments from $\mathcal{D}_{\text{HL}}$ (unroll length $U$).
21:        Sample a batch of primitive transitions from $\mathcal{D}_{\text{LL}}$.
22:        Compute policy/value/reward losses for the planner heads.
23:        Compute actor–critic losses for the budget policy $b_\psi(\cdot \mid h, z)$.
24:        Compute $\mathcal{L}_{\text{unified}}$ from Equation (3) (Appendix A.1).
25:        Update HL networks $(h_\theta, f_\theta, g_\theta, m_\phi, b_\psi)$ by minimizing the combined HL objective.
26:     **end if**
27:     **// Phase 3: Low-level updates**
28:     **if** $|\mathcal{D}_{\text{LL}}|$ is large enough **then**
29:        Sample a batch of primitive transitions from $\mathcal{D}_{\text{LL}}$.
30:        Update LL policy $\pi^L$ by minimizing the DQN loss $\mathcal{L}_{LL}$.
31:     **end if**
32: **end for**

---

penalties when the daily limit is exceeded), and evolves unvisited profits according to the same stochastic process as Feng et al. (2025). The per-decision reward is the aggregated profit collected that day minus any travel penalties.

**Low-level termination (both environments).** At each high-level decision the low-level controller selects primitive actions sequentially up to the allocated budget $b_t$, subject to illegal-action masking. Execution terminates early if the legal-action set becomes empty under the environment's feasibility constraints (*e.g.*, all selectable nodes are masked in AIM, or no feasible next city exists under the current day's constraints in SOP). Thus the realized duration is a stopping time $\tau(s, z, b) \leq b$ consistent with the definition used in Appendix A.

# E. Baseline Details

This section provides full baseline definitions and implementation notes referenced in Section 5. Unless otherwise stated, all learning-based baselines use the same environment instances, training/evaluation protocol, and number of seeds as LMTA.

## E.1. Adaptive Influence Maximization (AIM) Baselines

Following prior SSCO work (Tong et al., 2020; Feng et al., 2025), we structure AIM baselines as `HL-budget policy −` `LL-node selection policy`, where the high level decides how many seeds to allocate per stage and the low level selects individual nodes sequentially subject to the remaining budget and feasibility masking.

**High-Level Budget Policies.** Let $K$ denote the total seed budget and $T$ the number of decision stages.

- `average`: divides the total budget $K$ evenly across the $T$ stages.
- `normal`: allocates the entire budget $K$ in the first stage.
- `static`: divides the horizon into cycles and allocates budget only at the start of each cycle.

**Low-Level Node Selection Policies.** Given a stage budget, the low-level policy selects one node at a time from the feasible set (with masking of already active/removed nodes).

- `degree`: greedy heuristic selecting nodes with the highest out-degree.
- `score`: greedy heuristic selecting nodes based on expected immediate influence, $s_v = \sum_{u \in \mathcal{N}_{\text{in}}(v)} p(u, v)$.

**Flat DQN (AIM).** A non-hierarchical RL baseline using a single GNN-based Q-network to select one node per primitive step until the total budget $K$ is exhausted. This baseline does not use temporal abstraction; it performs standard deep Q-learning with illegal-action masking.

**WS-option (AIM).** The closest SSCO baseline (Feng et al., 2025), which learns a model-free high-level budgeting policy and an option-style low-level controller. WS-option communicates a scalar budget signal to the low level but does not perform model-based lookahead at the high level.

## E.2. Stochastic Orienteering Problem (SOP) Baselines

SOP baselines include both classical optimization heuristics and learning-based agents. All methods operate under the same episode horizon, daily constraints, and stochastic profit dynamics.

**Greedy Heuristic (SOP).** A domain-specific myopic policy that greedily selects the highest-profit node reachable within the daily distance limit (with penalties applied according to the environment rules when constraints are exceeded).

**Genetic Algorithm (GA) (SOP).** A meta-heuristic baseline that optimizes a fixed sequence of daily budget allocations over the entire horizon, following standard GA operators (selection, mutation, and crossover) and evaluating fitness under the SOP simulator.

**Flat DQN (SOP).** A non-hierarchical RL baseline analogous to AIM, selecting one routing decision (next city) at each primitive step using a GNN-based Q-network. It does not use subgoals or budgeted macro-actions.

**WS-option (SOP).** The SSCO option baseline (Feng et al., 2025) applied to SOP, where the high level is model-free and primarily controls budget allocation, while the low level learns a conditional policy for primitive selections.

### E.3. Additional Model-Based Baselines

In addition to SSCO-specific baselines, we include two general-purpose model-based RL baselines.

**Director** (Hafner et al., 2022). A goal-conditioned hierarchical model-based RL method with a fixed high-level stride $H$. To adapt Director to SSCO where budgets are variable and discrete, we use an even budget allocation strategy (fixed budget per stage) so that the effective high-level stride is fixed; this highlights the limitation of fixed-stride temporal abstraction under intertemporal resource constraints.

**Flat MuZero** (Schrittwieser et al., 2020). A non-hierarchical latent-space tree search baseline that operates directly on primitive actions using the same GNN encoder family as LMTA. This baseline provides a strong model-based comparison without explicit temporal abstraction, but incurs substantially higher branching and planning cost in combinatorial action spaces.

**Implementation notes (common).** All learning-based baselines use the same random seeds per configuration, report mean $\pm$ s.e.m., and are evaluated on held-out instances using the same protocol as LMTA. Where applicable, we apply the same illegal-action masking and action-feasibility constraints as the environments.

### E.4. Baseline Tuning Protocol

All baselines were run under the same AIM/SOP benchmark protocol as LMTA, using the same graph scale, horizons, resource budgets, random seeds, and evaluation procedure. For the large-scale comparisons, we re-checked the main scale- and compute-sensitive settings of the strongest baselines. For WS-option, this included the high-level resource-allocation policy and the relative training capacity of the high- and low-level learners. For Director, we checked the high-level decision stride. For MuZero-style baselines, we checked the number of MCTS simulations and, for Sampled MuZero, the number of sampled actions. For the domain heuristics, we checked the main resource-allocation and sampling settings used by the AIM and SOP implementations. In addition, for the compute–performance analysis, we swept the main test-time computation knob of each planning method and compared reward at matched decision latency. Reported test statistics were computed only after the baseline configurations were fixed.

## F. Network Architecture Details

Our framework is composed of two main hierarchical levels: the high-level planner and the low-level policy. Their architectures are detailed below.

**High-Level Networks.** The high-level planner is implemented with four functional modules in latent space, matching the interfaces in Section 3 and Section 4.2: representation $h_\theta$, dynamics $g_\theta^h$ with a reward head $R_\theta$, prediction $f_\theta$ (policy/value), and the separate budget policy $b_\psi$.

**Representation ($h_\theta$)** A two-layer message-passing Graph Neural Network (GNN) using linear aggregation with the graph adjacency matrix. The GNN is followed by ReLU activations, graph-level mean pooling, and a final linear projection to the $d$-dimensional latent space. The output is $\ell_2$-normalized on every forward pass to produce $\bar{h}_\theta(s)$.

**Dynamics ($g_\theta^h$) and macro-reward head ($R_\theta$)** A Multi-Layer Perceptron (MLP) takes the concatenation of the normalized latent state and the subgoal embedding, $[\bar{h}_\theta(s); z]$, and outputs the subsequent latent state $g_\theta^h(\bar{h}_\theta(s), z)$, which is also $\ell_2$-normalized. In addition, the macro-reward head $R_\theta$ is conditioned on the same $[\bar{h}_\theta(s); z]$ and predicts the cumulative macro reward associated with executing subgoal $z$ (implemented as a categorical support, unless stated otherwise).

**Prediction ($f_\theta$)** This module maps a latent state to a policy prior and value:

- **Policy Head:** An MLP predicting a categorical distribution over the discrete set of subgoals $\mathcal{Z}$.
- **Value Head:** An MLP predicting the scalar expected return from the current state.

**Budget policy** ($b_\psi$) An MLP conditioned on $[\bar{h}_\theta(s); z]$ that outputs a discrete distribution over possible budget allocations. This head is trained via the actor–critic objective described in Section 4 (budget details) and Appendix G.

**Low-Level Policy** ($\pi^L$). The low-level policy is implemented as a subgoal-conditioned GNN-based Q-network. It processes the graph state and computes Q-values for each primitive action (e.g., selecting a node). The GNN's node representations are concatenated with the current high-level subgoal embedding $z$ to make the policy goal-aware. The network employs illegal-action masking to ensure that only valid actions are selected during execution.

## F.1. Training Hyperparameters

*Table 4.* Key training hyperparameters (shared across AIM and SOP unless noted).

| Component | Hyperparameter | Symbol | Value |
|---|---|---|---|
| Latent Space | Latent dimension | $d$ | 128 |
| MuZero-style HL Planner | Learned subgoals | $\|\mathcal{Z}\|$ | 32 |
| | Unroll steps for training | $U$ | 5 |
| | Decision-time discount (tree-search backups) | $\gamma$ | 0.997 |
| MTS–SMDP World Model (Unified Loss) | LL per-step cap (unit sphere) | $\kappa$ | 0.10 |
| | Margin regularizer (L2) | $\lambda_m$ | $1 \times 10^{-4}$ |
| | Loss weight (cap / Pull) | $w_{\mathrm{cap}}$ | 0.10 |
| | Loss weight (push / margin) | $w_{\mathrm{push}}$ | 0.10 |
| | Loss weight (order / pairwise) | $w_{\mathrm{order}}$ | 0.10 |
| Low-Level (LL) Controller | $\varepsilon$-greedy schedule | $\varepsilon$ | $0.90 \to 0.05$ (exp., decay 0.995) |
| | Discount factor (primitive-time) | $\gamma_{\mathrm{LL}}$ | 0.997 |
| Optimization / Replay | Optimizer (all nets) | – | Adam |
| | HL learning rate (repr/dyn/pred/margin/budget) | $\eta_{\mathrm{HL}}$ | $1 \times 10^{-3}$ |
| | LL learning rate | $\eta_{\mathrm{LL}}$ | *see Tab. 5* |
| | Weight decay (all) | – | $1 \times 10^{-5}$ |
| | Gradient clipping | – | global norm 5.0 |
| | Batch size (HL / LL) | – | 8 (games/unrolls) / 8 (transitions) |
| Buffers / Data | HL replay buffer size (games) | – | 1,000 |
| | LL replay buffer size (steps) | – | 10,000 |
| | Seeds per configuration | – | 10 |
| | Evaluation metric | – | mean $\pm$ s.e.m. |

*Table 5.* Domain-specific overrides. Values omitted here inherit from Table 4.

| Domain | Hyperparameter | Symbol | AIM | SOP |
|---|---|---|---|---|
| HL Planner | MCTS simulations / HL decision | $N_{\mathrm{sim}}$ | 150 | 250 |
| HL Planner | PUCT exploration constant (init) | $c_{\mathrm{init}}$ | 2.5 | 1.5 |
| Exploration | MCTS root noise (training) | – | Dirichlet $\alpha = 0.30$, mix $\epsilon = 0.30$ | Dirichlet $\alpha = 0.20$, mix $\epsilon = 0.20$ |
| Training | Search temperature (training) | $\tau_{\mathrm{search}}$ | 1.0 | 1.5 |
| Optimization | LL learning rate | $\eta_{\mathrm{LL}}$ | $1 \times 10^{-3}$ | $5 \times 10^{-4}$ |
| Budget AC | Entropy bonus | $\beta$ | 0.01 | 0.02 |

# G. Training Details for the Budget Actor–Critic

**Budget action masking for shrinking feasible sets.** At HL step $k$, the feasible budget set depends on the remaining resources $B_k$, i.e., $b_k \in \{0, \ldots, B_k\}$. We implement the budget policy over a fixed universe $\{0, \ldots, B_{\max}\}$ but enforce feasibility by *logit masking* followed by renormalization. Specifically, the network outputs unnormalized logits $\ell_\psi(h_k, z_k) \in \mathbb{R}^{B_{\max}+1}$. We construct masked logits

$$\tilde{\ell}_\psi(b \mid h_k, z_k, B_k) = \begin{cases} \ell_\psi(b \mid h_k, z_k), & 0 \le b \le B_k, \\ -\infty, & b > B_k, \end{cases}$$

and define the feasible budget distribution by a masked softmax:

$$b_\psi(b \mid h_k, z_k, B_k) = \frac{\exp(\tilde{\ell}_\psi(b \mid h_k, z_k, B_k))}{\sum_{b'=0}^{B_k} \exp(\tilde{\ell}_\psi(b' \mid h_k, z_k, B_k))}.$$

In practice we implement $-\infty$ as a large negative constant (e.g., $-10^9$) for numerical stability. We store $B_k$ in the HL transition so that the same masking is applied consistently during off-policy training updates. At each high-level decision $k$, the planner outputs $(s_k, z_k)$; we sample $b_k \sim b_\psi(\cdot \mid h_k, z_k)$, roll out the low-level controller for up to $b_k$ primitive steps, and observe the realized macro reward $R_k = \sum_{i=0}^{\tau_k - 1} \gamma_{\mathrm{LL}}^i r_{t_k+i}$ and the next decision-time state $s_{k+1}$.

Because tree search branches only over subgoals and is budget-marginalized, the budget actor is trained using a target that conditions on the realized transition under the sampled $b_k$. Specifically, after the environment transitions to $s_{k+1}$, we run MCTS at $s_{k+1}$ and obtain the search-backed root value $v^{\mathrm{TS}}(s_{k+1})$. We then define the budget-policy return target

$$\hat{G}_k^{(b)} \;=\; R_k + \gamma\, v^{\mathrm{TS}}(s_{k+1}).$$

The critic $V_\eta(s, z)$ minimizes $\mathbb{E}\big[(\hat{G}_k^{(b)} - V_\eta(s_k, z_k))^2\big]$, and the actor maximizes $\mathbb{E}\big[\log b_\psi(b_k \mid s_k, z_k)\,(\hat{G}_k^{(b)} - V_\eta(s_k, z_k)) + \beta\,\mathcal{H}\big]$. We interleave actor–critic updates with representation/dynamics losses (including Equation (3)).

**Target network for the low-level critic.** To stabilize the low-level Q-learning updates, we maintain a separate target network $Q_{\mathrm{LL}}^{\mathrm{tgt}}$ initialized as a copy of the online low-level network $Q_{\mathrm{LL}}$. We use periodic *hard updates*: every $F$ low-level gradient steps, we set $Q_{\mathrm{LL}}^{\mathrm{tgt}} \leftarrow Q_{\mathrm{LL}}$. In our experiments, $F{=}100$ for AIM and $F{=}200$ for SOP (see Table 5).

For stability, we use a brief warm-up of three epochs before enabling the learned budget head $b_\psi(\cdot \mid s, z)$. During this warm-up the high-level planner still selects subgoals via MCTS and *all* high-level networks ($h_\theta, g_\theta, f_\theta$) and the low-level controller continue training from self-play, but the budget is sampled from a simple heuristic (average allocation). This delays credit assignment to the budget actor for just a few epochs, which empirically reduces early high-variance updates and avoids coupling instabilities between subgoal search and budget allocation, while *preserving* end-to-end gradients and on-policy targets for the world model from the outset. Conceptually, this is a lightweight alternative to the *wake–sleep* schedule in WS-option (Feng et al., 2025), where the "sleep" stage freezes the high-level budget policy and uses an average allocation so the low level can approach convergence before joint training resumes (and the high level is trained with MC targets to mitigate error propagation). In contrast, our three-epoch warm-up does *not* freeze any heads and does not pause high-level learning; it simply postpones learning of $b_\psi$ to dampen early interference, after which actor–critic training of the budget head is enabled and proceeds jointly with the rest of the system.

### G.1. Combined HL Objective, Loss Weights, and Stop-Gradient Choices

This section specifies the exact combined high-level objective minimized in Algorithm 1. A training batch samples an HL unroll segment of length $U$ from the HL replay, providing decision-time states $\{s_{k+u}\}_{u=0}^{U}$, executed subgoals $\{z_{k+u}\}_{u=0}^{U-1}$, realized macro rewards $\{R_{k+u}\}_{u=0}^{U-1}$, and search-backed targets $\{(\pi_{k+u}^{\mathrm{TS}}, v_{k+u}^{\mathrm{TS}})\}_{u=0}^{U}$ (stored during self-play).

**MuZero-style unroll and prediction losses.** We form latent unrolls as in MuZero: $h^0 = h_\theta(s_k)$ and $h^{u+1} = g_\theta^h(h^u, z_{k+u})$ for $u = 0, \ldots, U-1$. At each depth $u$, the prediction head outputs $(\mathbf{p}^u, v^u) = f_\theta(h^u)$. The dynamics model outputs the next latent and reward: $(h^{u+1}, \hat{R}^u) = g_\theta(h^u, z_{k+u})$. We minimize

$$\mathcal{L}_{\mathrm{MuZero}} = \sum_{u=0}^{U} \Big( w_\pi\, \mathrm{CE}\big(\mathbf{p}^u, \pi_{k+u}^{\mathrm{TS}}\big) + w_V\, \big(v^u - v_{k+u}^{\mathrm{TS}}\big)^2 \Big) + \sum_{u=0}^{U-1} w_R\, \ell_R\big(\hat{R}^u, R_{k+u}\big), \tag{10}$$

where $\mathrm{CE}(\cdot, \cdot)$ is cross-entropy on subgoals, and $\ell_R$ is the MuZero-style reward loss on the chosen reward parametrization (categorical support in our implementation; MSE if using a scalar reward head).

**MTS–SMDP unified loss.** We add the geometric constraints via $\mathcal{L}_{\mathrm{unified}}(\theta, \phi)$ from Equation (3), which itself is a weighted sum of Pull/Push/Order terms and includes the margin regularizer $\lambda_m \|m_\phi\|_2^2$.

**Budget actor–critic loss.** We include the budget actor–critic objective from Sec. 4.3 (written here as a minimization). Using the search-backed, transition-conditioned target defined in Sec. 4.3, $\hat{G}_k^{(b)} = R_k + \gamma v^{\mathrm{TS}}(s_{k+1})$, we minimize the critic loss and maximize the actor objective (equivalently minimize its negative):

$$\mathcal{L}_{\mathrm{budget}} = \mathbb{E}\left[\sum_k \left(w_{\mathrm{bc}}\,(V_\eta(h_k, z_k) - \hat{G}_k^{(b)})^2 + w_{\mathrm{ba}}\big(-\log b_\psi(b_k \mid h_k, z_k)\,\mathrm{sg}(A_k)\big) - w_{\mathrm{ent}}\,\beta\,\mathcal{H}(b_\psi(\cdot \mid h_k, z_k))\right)\right],$$
(11)

$$A_k = \hat{G}_k^{(b)} - V_\eta(h_k, z_k).$$

where $\mathrm{sg}(\cdot)$ denotes stop-gradient and $\mathcal{H}$ is entropy.

**Full combined HL objective.** The combined objective minimized for HL parameters $\Theta_{\mathrm{HL}} = \{\theta, \phi, \psi, \eta\}$ is

$$\mathcal{L}_{\mathrm{HL}} = \mathcal{L}_{\mathrm{MuZero}} + \lambda_{\mathrm{mts}}\,\mathcal{L}_{\mathrm{unified}} + \lambda_{\mathrm{bud}}\,\mathcal{L}_{\mathrm{budget}}.$$
(12)

**Loss weights (for replication).** Unless otherwise stated, we use unit weights for the standard MuZero terms ($w_\pi = w_V = w_R = 1$) and set $\lambda_{\mathrm{mts}} = \lambda_{\mathrm{bud}} = 1$. Within $\mathcal{L}_{\mathrm{unified}}$, we use the Pull/Push/Order weights reported in Appendix A.1 / Table 4 (default $w_{\mathrm{cap}} = w_{\mathrm{push}} = w_{\mathrm{order}} = 0.10$) and margin regularizer $\lambda_m$ as in Table 4. Budget-specific weights are $w_{\mathrm{bc}} = w_{\mathrm{ba}} = 1$ and $w_{\mathrm{ent}} = 1$, with entropy coefficient $\beta$ as reported in Table 5.

**Stop-gradient and anti-collapse mechanisms.** We do not backpropagate through MCTS: search targets ($\pi^{\mathrm{TS}}, v^{\mathrm{TS}}$) are stored in replay and treated as constants. In the budget actor–critic, we apply stop-gradient to the advantage $\mathrm{sg}(A_k)$ in Equation (11). We do not apply additional stop-gradient between the shared encoder and the MuZero/MTS losses; empirically, representation collapse is avoided by (i) supervised MuZero prediction targets (policy/value/reward), (ii) unit-sphere normalization of latents, and (iii) the Push/Order terms with a regularized margin head (Appendix A).

## H. Additional Ablations on MTS Loss Terms

We isolate the three components of the unified MTS–SMDP objective in Equation (3) and report average reward (mean $\pm$ s.e.m.) with $N=500$, $K=70$, and 10 seeds: (i) **Pull** = LL per-step cap (local geometric smoothness), (ii) **Push** = HL under-displacement margin (prevents vanishing macro steps), (iii) **Order** = budget-aware pairwise hinge (rank-calibrates durations). Each ablated variant is trained from scratch with the same protocol as the main experiments.

*Table 6.* AIM and SOP ($N=500$, $K=70$): Per-term loss ablation. Average reward $\pm$ s.e.m. (10 seeds).

| Variant | AIM: Avg. Reward | SOP: Avg. Reward |
|---|---|---|
| **Full (Ours)** | **324.15** $\pm$**2.38** | **31.60** $\pm$**1.94** |
| w/o **Pull** (LL cap) | 318.72 $\pm$3.20 | 24.66 $\pm$1.55 |
| w/o **Push** (margin) | 319.61 $\pm$2.95 | 25.14 $\pm$1.48 |
| w/o **Order** (pairwise hinge) | 320.17 $\pm$2.92 | 26.48 $\pm$1.52 |

As shown in Table 6, removing any single term reduces average reward, with the **LL cap (Pull)** being most critical (largest drop), followed by the **margin (Push)**, and then the **pairwise Order** term. This ordering is pronounced on SOP, where local geometric control and duration-aware macro steps are essential for routing under uncertainty. These trends align with our analysis: the cap establishes macro–micro coupling (Lem. A.9), the margin provides a duration-sensitive lower bound on macro steps (Prop. A.10), and the budget-aware pairwise hinge refines duration calibration (Thm. A.11).

## I. Hyper-parameter Sensitivity

We investigate four knobs: number of learned subgoals $|\mathcal{Z}|$, MCTS simulations per HL decision $N_{\mathrm{sim}}$, unroll steps $U$, and the LL per-step cap $\kappa$.

**Grids.** $|\mathcal{Z}| \in \{16, 32, 64\}$, $N_{\mathrm{sim}}^{\mathrm{AIM}} \in \{50, 100, 150, 200\}$, $N_{\mathrm{sim}}^{\mathrm{SOP}} \in \{150, 200, 250, 300\}$, $U \in \{3, 5, 7\}$, $\kappa \in \{0.05, 0.10, 0.15\}$. For each setting we keep total wall-clock within a budget by proportionally adjusting batch sizes. Results are reported in Table 7, Tables 8–9, Table 10, and Table 11.

*Table 7.* Sensitivity to number of learned subgoals $|\mathcal{Z}|$ (10 seeds). Means $\pm$ s.e.m. are reported.

| $|\mathcal{Z}|$ | 16 | 32 | 64 |
|---|---|---|---|
| AIM | 320.85 $\pm$3.25 | 324.15 $\pm$2.38 | 323.89 $\pm$2.56 |
| SOP | 29.13 $\pm$1.16 | 31.60 $\pm$1.94 | 31.52 $\pm$1.39 |

*Table 8.* AIM: Sensitivity to MCTS simulations $N_{\mathrm{sim}}$ (10 seeds). Means $\pm$ s.e.m. are reported.

| $N_{\mathrm{sim}}$ | 50 | 100 | 150 | 200 |
|---|---|---|---|---|
| AIM | 322.05 $\pm$3.88 | 323.92 $\pm$2.45 | 324.15 $\pm$2.38 | 324.03 $\pm$2.21 |

*Remark (cap sensitivity).* The LL cap sets the per-step latent-motion scale that enters the theoretical coupling; performance varies within $\sim$1 s.e.m. across the sweep, suggesting low sensitivity once the scale is reasonable.

**Default hyperparameter choice.** Across AIM and SOP, we use $w_{\mathrm{cap}} = w_{\mathrm{push}} = w_{\mathrm{order}} = 0.10$, $\kappa = 0.10$, $\lambda_m = 10^{-4}$, and $|\mathcal{Z}| = 32$. The cap $\kappa$ is chosen to keep one primitive transition local in latent space. The three MTS loss weights are set equally to balance the Pull, Push, and Order terms.

## J. Zero-Shot Generalization to Large Graphs

Tables 12 and 13 report large-graph generalization for agents trained once on $N{=}100$ graphs with $(T, K){=}(10, 60)$ and evaluated *zero-shot* on unseen graphs with $N \in \{1000, 1500, 2000, 2500\}$, using identical settings.

In AIM, LMTA consistently outperforms all baselines, with the margin increasing as $N$ grows; degree/score heuristics remain competitive only at smaller $N$, while WS-option and Flat DQN degrade with action-space scale and propagation depth. In SOP, LMTA is the only learning-based method that reliably surpasses the strong `Greedy` heuristic on the largest graphs, whereas WS-option and Flat DQN struggle to maintain globally coherent routes as instance size increases. We attribute this robustness to macro-level MuZero-style planning combined with a multi-timescale latent geometry that enforces local smoothness yet preserves semantically large subgoal displacements, enabling temporally calibrated, single-step HL evaluations that transfer without re-tuning.

## K. Qualitative Analysis of Subgoal Specialization

Beyond quantitative metrics, we seek to understand if the learned subgoals correspond to qualitatively distinct and strategically meaningful behaviors. To investigate this, we visualize the ideal actions a specialized low-level policy would take when conditioned on different subgoals in both the AIM and SOP domains, as shown in Figure 3. In both visualizations, the size of a node is proportional to its intrinsic value in the problem: its degree in the AIM task and its profit in the SOP task.

Figure 3a illustrates this for the AIM task on an exemplar graph. The visualization shows a clear strategic divergence: one subgoal (red) learns to target the central, high-degree hubs for broad, high-impact influence, which are visibly the largest nodes. In contrast, the other subgoal (blue) focuses on a dense local community of smaller nodes, representing a more focused, saturating strategy. Similarly, Figure 3b shows the emergent strategies for the SOP task. One subgoal (red) identifies high-profit targets scattered across the map, which are clearly depicted as larger nodes, while the second subgoal (blue) identifies a dense cluster of smaller, lower-profit nodes, prioritizing travel efficiency. Together, these visualizations demonstrate that LMTA's abstract subgoals can correspond to concrete, interpretable, and strategically diverse plans.

Although these visualizations indicate strategically distinct behaviors, the subgoals remain learned latent abstractions rather than a human-authored control vocabulary. Improving their interpretability and controllability is an important direction for human-in-the-loop deployment.

*Table 9.* SOP: Sensitivity to MCTS simulations $N_{\text{sim}}$ (10 seeds). Means $\pm$ s.e.m. are reported.

| $N_{\text{sim}}$ | 150 | 200 | 250 | 300 |
|---|---|---|---|---|
| SOP | 30.91 $\pm$2.18 | 31.27 $\pm$1.25 | 31.60 $\pm$1.94 | 31.44 $\pm$1.36 |

*Table 10.* Sensitivity to unroll steps $U$ (10 seeds). Means $\pm$ s.e.m. are reported.

| $U$ | 3 | 5 | 7 |
|---|---|---|---|
| AIM | 322.93 $\pm$3.95 | 324.15 $\pm$2.38 | 324.02 $\pm$2.72 |
| SOP | 29.91 $\pm$2.21 | 31.60 $\pm$1.94 | 31.45 $\pm$2.28 |

## L. Validation of the Multi-Timescale World Model via Kendall's Tau

We empirically investigate whether the Multi-Timescale SMDP (MTS) objective enables the agent to learn a temporally meaningful latent space, where the model's internal score for a subgoal, $\sigma_{\theta,\phi}(s, z)$, is monotonically related to the subgoal's true duration, $\hat{\tau}(s, z)$. To this end, we use Kendall's Rank Correlation Coefficient ($\tau$). A high positive correlation ($\tau \to 1$) indicates that the model has successfully learned the relative temporal ordering of its abstract actions, a crucial capability for effective long-term planning.

### L.1. Order-Consistency of the MTS Objective

We analyze the quality of the learned temporal ordering for the two variants of our framework that incorporate the MTS objective. Table 14 presents the Kendall's Tau correlation for both the fixed-margin version (Variant D) and the full LMTA model with a learnable margin (Variant E). The results provide clear, quantitative evidence that the MTS objective is highly effective. Both variants achieve a strong positive correlation, with Tau values consistently in the $0.69 \sim 0.76$ range. This indicates that the model has successfully learned a reliable sense of temporal order. Furthermore, the full LMTA model (E) consistently achieves a higher correlation than the fixed-margin version (D), demonstrating the benefit of the learnable, subgoal-specific margin in further refining the temporal representation.

### L.2. Calculation Methodology

The reported Kendall's Tau values were generated following a procedure for each of the 10 random seeds per algorithm variant.

1. **Model Selection:** We take the final trained agent from each of the 10 independent runs.

2. **Data Sampling:** We perform rollouts on a held-out set of 50 test environment instances. During these rollouts, we sample 100 high-level states $s$ at random.

3. **Generating Paired Rankings:** For each sampled state $s$, we generate two ranked lists over the entire set of available subgoals $\mathcal{Z} = \{z_1, z_2, \ldots, z_{|\mathcal{Z}|}\}$:

   - **Learned Score Ranking:** We perform a forward pass for each subgoal to compute the list of learned scores $\{f(s, z_1), f(s, z_2), \ldots, f(s, z_{|\mathcal{Z}|})\}$.
   - **Empirical Duration Ranking:** To get a stable estimate of the ground-truth duration, we execute each subgoal $z_i$ from state $s$ five separate times, allowing the low-level policy to run until termination. We record the number of steps taken in each of the five runs and use the average duration to form the list $\{\hat{\tau}(s, z_1), \hat{\tau}(s, z_2), \ldots, \hat{\tau}(s, z_{|\mathcal{Z}|})\}$. This averaging mitigates the effects of stochasticity in both the environment and the low-level policy.

4. **Calculation and Aggregation:** For each of the 100 sampled states, we compute the Kendall's Tau correlation coefficient between the score ranking and the duration ranking. The final Tau value for a single seed is the average of these 100 individual Tau calculations.

5. **Final Reporting:** The values presented in the tables are the mean and standard error of the mean (s.e.m.) of the final Tau values from the 10 independent seeds.

*Table 11.* Sensitivity to the LL cap $\kappa$ (10 seeds). Means $\pm$ s.e.m. are reported.

| $\kappa$ | 0.05 | 0.10 | 0.15 |
|---|---|---|---|
| AIM: Avg. Reward | 324.01 $\pm$2.86 | 324.15 $\pm$2.38 | 323.94 $\pm$2.64 |
| SOP: Avg. Reward | 31.32 $\pm$1.82 | 31.60 $\pm$1.94 | 31.41 $\pm$1.85 |

*Table 12.* AIM: Generalization to larger graphs (trained on graph $N = 100$, $T = 10$, $K = 60$). All improvements of LMTA over the best baseline are statistically significant (p-value $\leq 0.05$).

| Method | $N = 1000$ | $N = 1500$ | $N = 2000$ | $N = 2500$ |
|---|---|---|---|---|
| **LMTA (Ours)** | **416.06** $\pm$5.81 | **540.78** $\pm$6.92 | **605.55** $\pm$7.54 | **654.29** $\pm$8.11 |
| WS-option | 370.15 $\pm$5.25 | 465.22 $\pm$6.13 | 510.98 $\pm$6.88 | 531.40 $\pm$7.21 |
| Flat DQN | 341.82 $\pm$5.40 | 431.65 $\pm$6.12 | 482.37 $\pm$6.78 | 501.93 $\pm$7.05 |
| average-degree | 366.36 $\pm$5.11 | 493.57 $\pm$6.45 | 549.02 $\pm$7.02 | 590.10 $\pm$7.63 |
| average-score | 387.80 $\pm$5.43 | 498.78 $\pm$6.51 | 572.56 $\pm$7.28 | 617.14 $\pm$7.95 |
| normal-degree | 166.01 $\pm$3.11 | 174.82 $\pm$3.28 | 180.55 $\pm$3.41 | 184.61 $\pm$3.52 |
| normal-score | 178.95 $\pm$3.35 | 189.42 $\pm$3.55 | 197.09 $\pm$3.71 | 201.95 $\pm$3.83 |
| static-degree | 355.20 $\pm$4.98 | 470.11 $\pm$6.21 | 521.73 $\pm$6.91 | 560.88 $\pm$7.44 |
| static-score | 371.44 $\pm$5.18 | 482.90 $\pm$6.33 | 550.16 $\pm$7.09 | 599.23 $\pm$7.77 |

This comprehensive process ensures that the reported correlation is a robust and reliable measure of the model's learned temporal awareness.

## M. Calibration of Value–Geometry Smoothness

We empirically examine the link between latent displacement and value variation posited by our smoothness assumptions. At each high-level decision point we form pairs $(u, v)$ with $u = h_\theta(s)$ and $v = g_\theta(h_\theta(s), z)$, and compute the chordal distance on the unit sphere, $d(u, v) = \|u - v\|_2$. We estimate $|\Delta V| \triangleq |V(u) - V(v)|$ using the same training targets as in the main method. Unless stated otherwise, results aggregate **5 seeds** over **25 epochs** per task (total $n = 125$ points) under the task settings $T{=}10$, $K{=}60$.

We report Kendall's $\tau_b$ and Spearman's $\rho$ between $d(u, v)$ and $|\Delta V|$, along with nonparametric bootstrap 95% confidence intervals. Both benchmarks show a clear positive monotonic association (Table 15); the corresponding scatter plots are in Fig. 4. The strong rank correlations imply that larger latent displacements are associated with larger absolute value changes, supporting the use of latent geometry as a proxy for temporal/strategic scale in SMDP planning.

## N. Additional Results

We report additional evaluations for model-based baselines, large-action-space planning, scaling with graph size, small-scale optimality calibration, and budget-aware search variants.

**Baselines.**

- **Director** (Hafner et al., 2022): A goal-conditioned hierarchical model-based RL agent. Since Director uses a fixed high-level stride $H$, we adapted it to SSCO by using an even budget allocation strategy (budget per stage is fixed).

- **Flat MuZero** (Schrittwieser et al., 2020): A standard model-based RL agent without hierarchy, operating directly on primitive actions using the same GNN encoder as LMTA.

- **Sampled MuZero** (Hubert et al., 2021): A MuZero-style baseline that samples candidate primitive actions during tree search to reduce branching in large discrete action spaces.

**Additional Task: Power-2500.** We evaluate all learning-based methods on the **Power-2500** task, which is based on the large-scale influence maximization instance from Feng et al. (2025) with graph size $N = 2500$, but with a significantly higher budget ($K = 60$).

*Table 13.* SOP: Generalization to larger graphs (trained on $N = 100$, $T = 10$, $K = 60$). All improvements of LMTA over the best baseline are statistically significant (p-value $\leq 0.05$).

| Method | $N = 1000$ | $N = 1500$ | $N = 2000$ | $N = 2500$ |
|---|---|---|---|---|
| **LMTA (Ours)** | **31.27** $\pm$**1.21** | **33.39** $\pm$**1.29** | **36.15** $\pm$**1.38** | **38.88** $\pm$**1.45** |
| WS-option | 18.63 $\pm$0.99 | 19.07 $\pm$1.03 | 20.52 $\pm$1.11 | 21.23 $\pm$1.15 |
| Flat DQN | 20.41 $\pm$1.06 | 22.18 $\pm$1.12 | 23.95 $\pm$1.18 | 25.72 $\pm$1.22 |
| Greedy | 23.12 $\pm$1.10 | 26.05 $\pm$1.19 | 27.37 $\pm$1.24 | 29.04 $\pm$1.30 |
| GA | 12.62 $\pm$0.85 | 10.96 $\pm$0.79 | 11.37 $\pm$0.81 | 11.77 $\pm$0.83 |

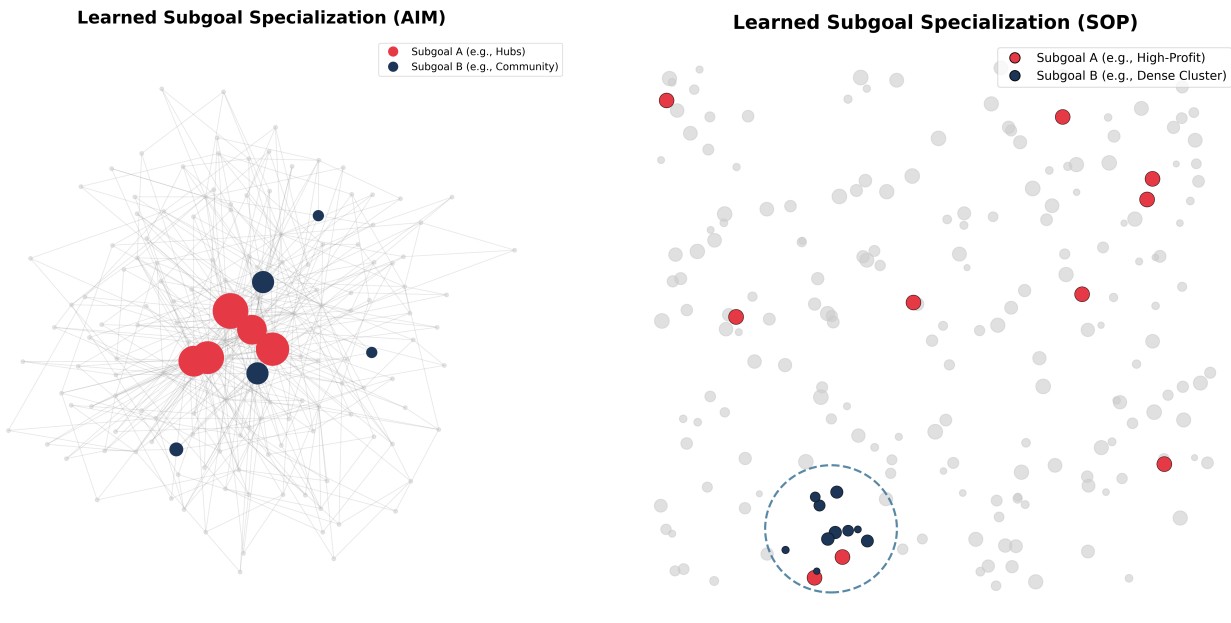

*(a)* AIM: Hub-Hunting vs. Community-Targeting      *(b)* SOP: High-Profit vs. Dense-Cluster

*Figure 3.* Visualization of learned subgoal specialization. In both domains, LMTA learns distinct and interpretable strategies. (a) In AIM, one subgoal targets high-degree central hubs (red) while another focuses on a dense community (blue). (b) In SOP, one subgoal prioritizes scattered high-profit nodes (red), whereas another concentrates on a spatially dense neighborhood (blue; dashed circle), illustrating region-level specialization.

**Quantitative Results.** Table 16 summarizes performance against additional learning-based baselines and on Power-2500. The results are consistent with the main AIM/SOP comparisons: LMTA obtains the highest mean return across the reported settings.

**Scaling with graph size.** Figure 6 reports AIM performance as a function of graph size for $T = 10, K = 30$. Both methods are retrained and evaluated under the same protocol at each graph size.

**Small-Scale Exact Solver Calibration.** Table 19 reports performance on tiny AIM instances where exact stochastic expectimax with memoization is tractable.

**Additional Learning Curves.** Figure 7 reports learning curves for the $K = 70$ AIM/SOP settings and the $K = 60$ Power-2500 benchmark.

**Budget Sampling in MCTS.** Table 20 reports performance for the default budget-marginalized model and variants that sample budgets as chance variables during MCTS.

*Table 14.* Kendall's Tau correlation between the learned score $\sigma_{\theta,\phi}(s,z)$ and empirical duration $\hat{\tau}(s,z)$ for the variants incorporating the MTS objective. Both achieve a strong positive correlation, directly validating the effectiveness of our proposed mechanism.

| Algorithm Variant | AIM Kendall's $\tau$ | SOP Kendall's $\tau$ |
|---|---|---|
| D: + MTS (Fixed Margin) | 0.71 $\pm$0.04 | 0.69 $\pm$0.05 |
| E: LMTA (Full Model) | 0.74 $\pm$0.03 | 0.76 $\pm$0.04 |

*Table 15.* Calibration of value–geometry smoothness. Rank correlations between chordal distance $d(u,v)$ and $|\Delta V|$ with bootstrap 95% CIs. Each task uses $n{=}125$ points (5 seeds $\times$ 25 epochs), $T{=}10$, $K{=}60$. These empirical rank relations complement Assumption A.6 and the monotone-ordering guarantees (Lemma A.9, Proposition A.10, and Theorem A.11), which underpin the regret bound in Theorem A.13.

| Task | Kendall's $\tau_b$ [95% CI] | Spearman's $\rho$ [95% CI] |
|---|---|---|
| AIM | **0.493** [0.401, 0.580] | **0.666** [0.536, 0.779] |
| SOP | **0.559** [0.475, 0.629] | **0.748** [0.634, 0.813] |

# O. Computational Cost

We provide a detailed breakdown of LMTA's computational cost in Table 22 and a comparison against baselines in Table 23.

For the AIM profiling setting ($T = 10, K = 70$), environment interaction counts are matched across methods: all methods use about 10 high-level decisions and 69–70 low-level actions/seeds per instance. The wall-clock differences therefore primarily reflect internal planning and model-evaluation compute.

## O.1. LMTA Detailed Metrics

In Table 22, the decision latency includes MCTS, subgoal selection, budget selection, and low-level action selection. Episode latency sums per-decision latency over $T$ decisions. CUDA synchronization is applied around timed regions.

## O.2. Comparison with Baselines

We compare the computational cost of LMTA against baselines in Table 23. Measurements were taken on the same hardware setup described above.

**Analysis.** LMTA has higher decision latency than WS-Option, Director, and Flat DQN because it performs MCTS over learned subgoals, while remaining below Flat MuZero, whose search operates over primitive actions. Figure 8 reports reward across test-time planning budgets. LMTA already obtains high reward in the greedy no-search setting and changes modestly as additional planning is added. This pattern is consistent with the training procedure: search-improved targets can be partly amortized into the learned high-level policy and value function, while online MCTS provides a smaller additional improvement at inference.

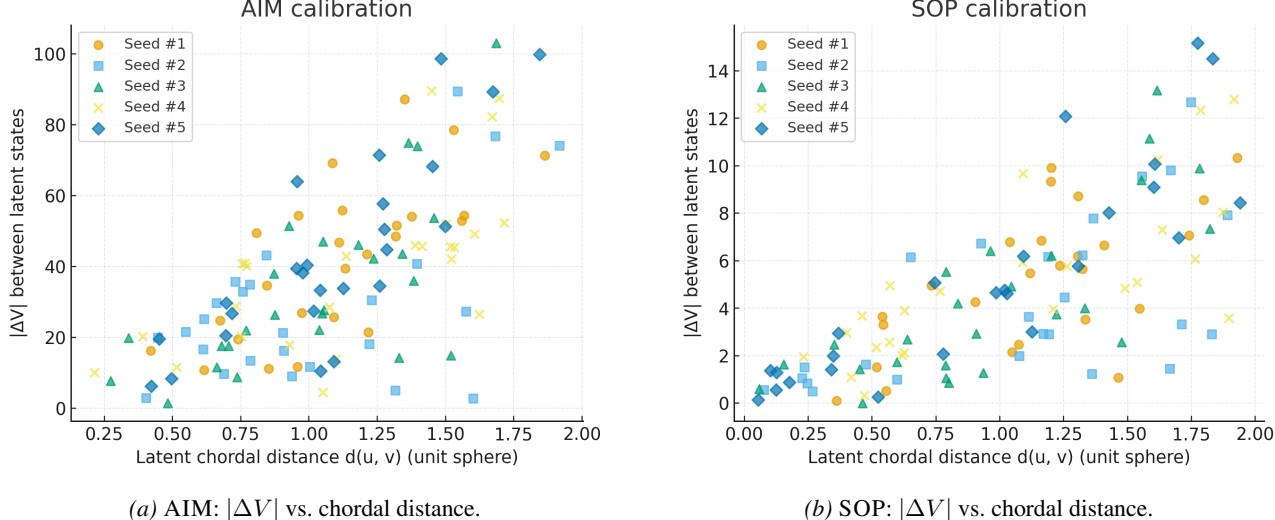

*(a)* AIM: $|\Delta V|$ vs. chordal distance.

*(b)* SOP: $|\Delta V|$ vs. chordal distance.

*Figure 4.* Calibration plots probing the value–geometry relationship. Points aggregate 5 seeds $\times$ 25 epochs per task.

*Table 16.* Comparison with additional learning-based baselines on AIM, SOP, and Power-2500. Mean $\pm$ s.e.m. over 10 seeds.

| Method | AIM ($N = 500, K = 70$) | SOP ($N = 500, K = 70$) | Power ($N = 2500, K = 60$) |
|---|---|---|---|
| **LMTA** | **324.15** $\pm$**2.38** | **31.60** $\pm$**1.94** | **856.14** $\pm$**10.20** |
| WS-Option | 301.53 $\pm$9.10 | 12.99 $\pm$1.25 | 676.86 $\pm$21.58 |
| Flat DQN | 243.46 $\pm$3.78 | 15.90 $\pm$0.67 | 464.78 $\pm$14.28 |
| Director | 306.53 $\pm$2.58 | 20.74 $\pm$0.93 | 725.61 $\pm$5.00 |
| Flat MuZero | 260.60 $\pm$8.41 | 17.75 $\pm$0.30 | 354.42 $\pm$4.32 |

*Table 17.* MuZero-style planning baselines on large-budget AIM and SOP settings. Mean $\pm$ s.e.m. over 10 seeds.

| Method | AIM ($N = 500, K = 70$) | SOP ($N = 500, K = 70$) |
|---|---|---|
| Flat MuZero | 260.60 $\pm$8.41 | 17.75 $\pm$0.30 |
| Sampled MuZero | 283.93 $\pm$2.55 | 19.41 $\pm$0.46 |
| **LMTA** | **324.15** $\pm$**2.38** | **31.60** $\pm$**1.94** |

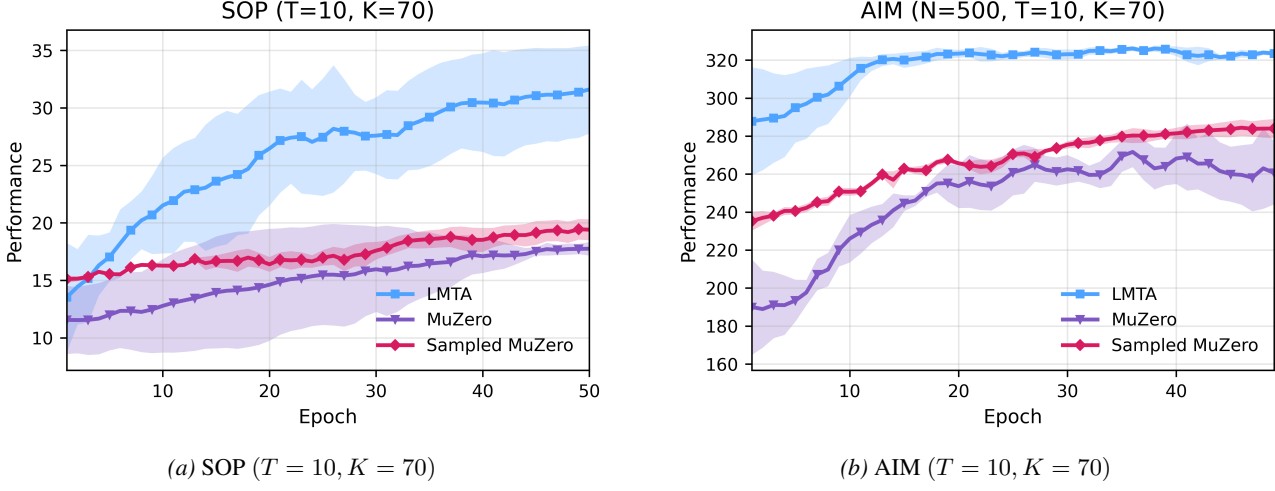

*(a)* SOP ($T = 10, K = 70$)

*(b)* AIM ($T = 10, K = 70$)

*Figure 5.* Training curves for LMTA and MuZero-style baselines on large-budget SOP and AIM settings.

*Table 18.* AIM results at $N = 200$ graph size. Mean $\pm$ s.e.m.

| Method | $T = 10, K = 10$ | $T = 10, K = 20$ | $T = 10, K = 30$ | $T = 20, K = 10$ |
|---|---|---|---|---|
| WS-Option | 78.84 $\pm$2.86 | 116.21 $\pm$3.71 | 128.63 $\pm$3.89 | 81.14 $\pm$2.93 |
| **LMTA** | **83.21** $\pm$**2.41** | **123.83** $\pm$**3.12** | **137.28** $\pm$**3.46** | **86.16** $\pm$**2.66** |

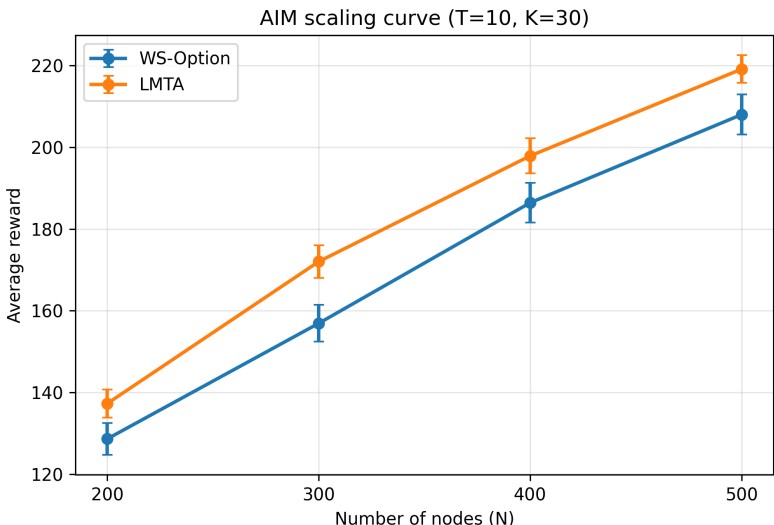

*Figure 6.* Scaling with graph size on AIM ($T = 10, K = 30$). Average reward is reported as a function of the number of nodes $N$. Both methods are retrained and evaluated under the same protocol at each graph size.

*Table 19.* Exact-solver calibration on tiny AIM instances ($N = 10, T = 4, K = 4$).

| Method | Value | Gap to optimum | Ratio to optimum |
|---|---|---|---|
| Exact optimum | **8.91** | 0.00 | **1.00** |
| LMTA | 8.73 | 0.18 | 0.98 |
| WS-Option | 7.72 | 1.19 | 0.87 |

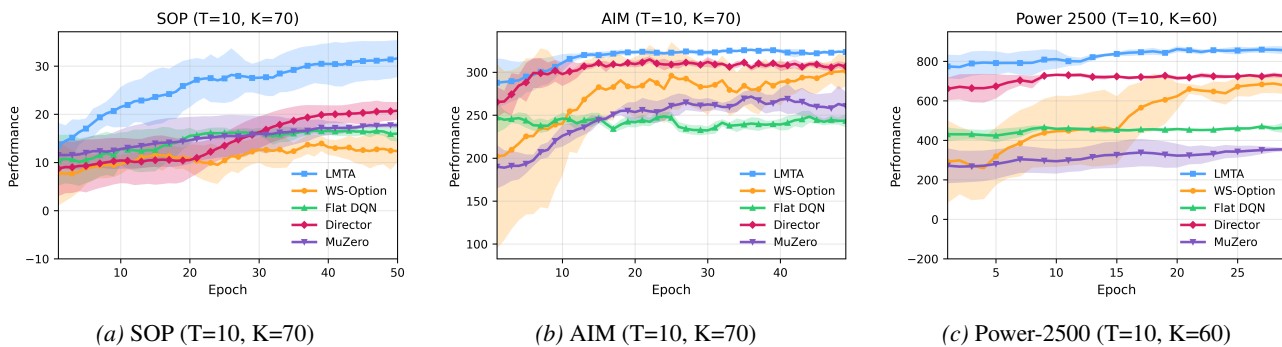

*(a)* SOP (T=10, K=70)  *(b)* AIM (T=10, K=70)  *(c)* Power-2500 (T=10, K=60)

*Figure 7.* Learning curves for $K = 70$ AIM/SOP settings and $K = 60$ Power-2500.

*Table 20.* Budget sampling variants in MCTS. Mean $\pm$ s.e.m. over 10 seeds.

| Method | AIM ($T = 10, K = 70$) | SOP ($T = 10, K = 70$) |
|---|---|---|
| **LMTA** | **324.15** $\pm$**2.38** | **31.60** $\pm$**1.94** |
| LMTA + chance-budget MCTS (3 samples) | 322.8 $\pm$2.93 | 29.5 $\pm$2.31 |
| LMTA + chance-budget MCTS (1 sample) | 321.9 $\pm$3.12 | 29.1 $\pm$2.45 |

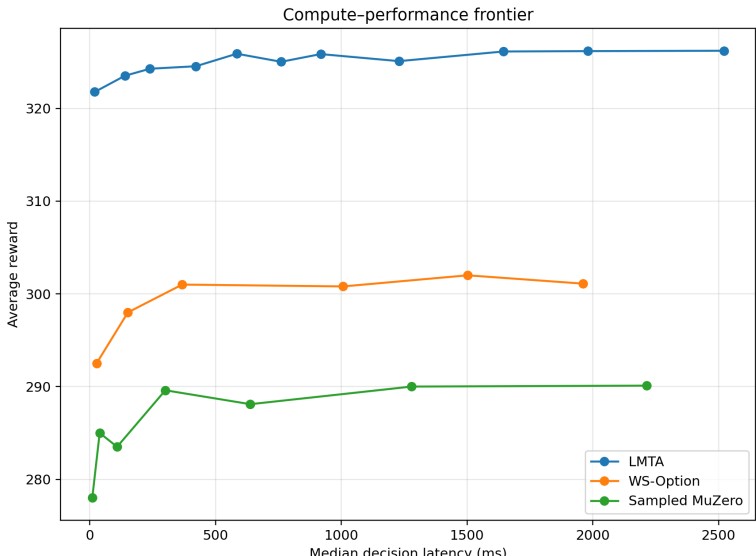

*Figure 8.* Compute–performance frontier on AIM. Average reward is reported against median decision latency under matched test-time compute sweeps.

*Table 21.* Setup for LMTA profiling.

| Item | Value |
|---|---|
| GPU | NVIDIA Tesla V100 (32 GB HBM2) |
| CUDA / cuDNN | 11.7 / 8.5 |
| PyTorch | 2.0.1 |
| Python | 3.8.19 |
| OS | Linux (kernel 4.15.0-135-generic) |
| Mixed precision (AMP) | disabled |
| Task | AIM with $N = 500$, $T = 10$, $K = 70$ |
| Subgoals / MCTS simulations | 32 / 150 |

*Table 22.* LMTA Results.

| Metric | Value |
|---|---|
| Training time / epoch | 62.26 s |
| Decision latency (median / p95) | 441.0 ms / 519.5 ms |
| Episode latency (mean; $T{=}10$) | 4,288.4 ms |
| Episodes / epoch | 16 |
| Env micro-steps (true frames) | 55,154 |

*Table 23.* Wall-clock time comparison (AIM, $N = 500$, $T = 10$, $K = 70$).

| Method | Decision Latency (ms) | Training Time / Epoch (s) |
|---|---|---|
| Flat DQN | 62.65 | 329.03 |
| WS-Option | 101.21 | 12.56 |
| Director | 40.38 | 22.5 |
| Flat MuZero | 851.53 | 822.39 |
| **LMTA** | 441.03 | 62.34 |

