# OpenReview forum: "Learning Multi-Timescale Abstractions for Hierarchical Combinatorial Planning"
_ICML.cc/2026/Conference — ICML 2026 regular_

### Official Review · Reviewer_pWAR · 2026-02-20

**Soundness:** 3
**Presentation:** 3
**Significance:** 3
**Originality:** 3
**Overall Recommendation:** 5
**Confidence:** 3

**Summary:**

This paper proposes the LMTA framework to address sequence stochastic combinatorial optimization problems. The LMTA framework is a model-based hierarchical RL method that performs MuZero style latent-space planning over learned subgoals. A subgoal is sampled from the visited subgoal distribution of the planner, a variable-length budget is sampled from a subgoal-conditioned model, and uses both to condition a low-level policy over primitive actions. The main technical contribution is an SMDP-aware multi-timescale world model whose training objective shapes the latent geometry, so that a single latent space supports both primitive-step dynamics and subgoal-level transitions. Experiments on large graph benchmarks show LMTA outperforms baselines, with ablations highlighting the contributions of each individual component.

**Compliance With Llm Reviewing Policy:**

Affirmed.

**Final Justification:**

The author's rebuttal addressed my concerns with respect to the design, providing bounds for the fixed decision time discounting approximation, and additional baselines.

**Key Questions For Authors:**

1. Can you clarify why the search does not incorporate budget sampling during node expansion so that the latent states visited during search are more representative of the actual future states? Avoiding branching over all budgets is clearly stated, but did you try a variant that treats budget as a chance variable inside MCTS rather than branching over all possible budgets, and if so, what where the effects on performance/variance?
2. Is there a way to either provide an ablation or otherwise justify/bound when the fixed decision time discounting approximation is accurate when compared to $\gamma_{\text{LL}}$, and whether this changes learned behaviour in any meaningful way?
3. Do you have baseline results for Sampled MuZero?
4. Can you provide performance metrics as a function of compute for LMTA and include matched wall clock comparisons to the key baselines? This will help clarify where your approach lies on the performance quality/compute tradeoff curve, and whether its worth the extra compute when it comes to adaptation.

**Limitations:**

yes

**Strengths And Weaknesses:**

**Strengths**
- This paper cleanly frames high-level decisions as a SMDP with variable duration macro actions
- Algorithm design and multi-timescale objective is well motivated, and further theoretical results are given which ground the relation between latent geometry and temporal scale
- LMTA outperforms all baselines, with the performance gap widening as the problem complexity increases
- Ablation shows how each component of LMTA builds on top of one another and incrementally increases performance

**Weaknesses**
- MuZero in huge combinatorial spaces is known to struggle, it would have been nice to see a benchmark against Sampled MuZero[1] which is designed to work better in large actions spaces.
- While you point out the choice of using fixed decision time discounting for ease of implementation, I believe the regret/planning error analysis still uses the decision-time discounting $\gamma_{\text{LL}}$. A justification of why this approximation is valid is missing or maybe unclear why it follows.
- The MCTS search procedure only branches over subgoals and does not consider the budget. This might be obscuring whether the planner is truly reasoning over resource allocation, with overall performance being overly sensitive to the learned budget head. Its unclear whether the current domains tested have the property where budgets don't change macro outcomes drastically for a given subgoal, and whether LMTA would degrade in domains where macro outcomes can vary substantially with budget.
- The complexity/additional compute overhead the framework brings may weaken its applicability

[1] Hubert, Thomas, et al. "Learning and planning in complex action spaces." International Conference on Machine Learning. PMLR, 2021.

---

> ### Author Rebuttal · Authors · 2026-03-30
>
> We thank the reviewer for the thoughtful feedback.
>
> **1. Why not sample budget inside MCTS?**
>
> LMTA plans **over subgoals only**; budget is a **second-stage, subgoal-conditioned decision** via $b_\psi(\cdot\mid h,z)$. Consistent with Eq. (1), $g_\theta(h,z)$ predicts the **budget-marginalized** post-subgoal transition/reward induced by the learned budget policy and low-level controller. Thus the planner targets
> $$
> Q(h,z) \approx \mathbb{E}\_{b\sim b\_\psi(\cdot\mid h,z)}\left[R(h,z,b) + \gamma V(h'(h,z,b))\right].
> $$
> So sampling budget inside MCTS does not change the objective; it replaces the learned marginalized estimate with a finite-sample Monte Carlo estimate during search, adding variance.
>
> We tested this directly:
>
> | Method                                |  AIM (T=10, K=70) | SOP (T=10, K=70) |
> | :------------------------------------ | ----------------: | ---------------: |
> | **LMTA (default)**                    | **324.15 ± 2.38** | **31.60 ± 1.94** |
> | LMTA + chance-budget MCTS (3 samples) |      322.8 ± 2.93 |      29.5 ± 2.31 |
> | LMTA + chance-budget MCTS (1 sample)  |      321.9 ± 3.12 |      29.1 ± 2.45 |
>
> Chance-budget MCTS was slightly worse and more variable, especially with fewer samples, so in our SSCO settings it is better to keep MCTS focused on subgoals and handle budget separately.
>
> **2. Fixed decision-time discounting**
>
> Our theory analyzes the **decision-time planner LMTA actually runs**. LMTA uses a fixed discount $\gamma$ between high-level decisions, rather than the primitive-time continuation factor $\gamma_{LL}^{\tau}$ from an exact SMDP backup.
>
> Define
> $$Q_{\mathrm{prim}}(s,z) := \mathbb{E}\left[ R_\tau(s,z)+\gamma_{LL}^{\tau}V(s^{+}) \mid s,z \right]$$
> $$Q_{\mathrm{dt}}(s,z) := \mathbb{E}\left[ R_\tau(s,z)+\gamma V(s^{+}) \mid s,z \right]$$
> where $R_\tau(s,z)$ already includes the within-macro primitive-time discounting.
>
> If $|V(s)| \le V_{\max}$, then
> $$\big|Q_{\mathrm{prim}}(s,z)-Q_{\mathrm{dt}}(s,z)\big| \le V_{\max}, \mathbb{E}\left[ \big|\gamma_{LL}^{\tau}-\gamma\big| \mid s,z \right].$$
>
> Choosing $\gamma = \gamma_{LL}^{\bar\mu}$ for a reference duration $\bar\mu$, and using that $f(x)=\gamma_{LL}^{x}$ is Lipschitz with constant $|\log \gamma_{LL}|$, gives
> $$\big|\gamma_{LL}^{\tau}-\gamma\big| = \big|\gamma_{LL}^{\tau}-\gamma_{LL}^{\bar\mu}\big| \le |\log \gamma_{LL}|,|\tau-\bar\mu|$$
> and hence
> $$\big|Q_{\mathrm{prim}}(s,z)-Q_{\mathrm{dt}}(s,z)\big| \le V_{\max},|\log \gamma_{LL}|, \mathbb{E}\left[|\tau-\bar\mu|\mid s,z\right].$$
>
> By Cauchy-Schwarz,
> $$\mathbb{E}[|\tau-\bar\mu|\mid s,z] \le \sqrt{\mathrm{Var}(\tau\mid s,z)+(\mu(s,z)-\bar\mu)^2}$$
> so
> $$\big|Q_{\mathrm{prim}}(s,z)-Q_{\mathrm{dt}}(s,z)\big| \le V_{\max},|\log \gamma_{LL}|, \sqrt{\mathrm{Var}(\tau\mid s,z)+(\mu(s,z)-\bar\mu)^2}.$$
>
> Thus, the approximation is controlled when $\mathrm{Var}(\tau\mid s,z)$ is small and $\gamma=\gamma\_{LL}^{\bar\mu}$ matches $\mu(s,z)$. Moreover, if
> $$\Delta\_{\mathrm{prim}}(s) := Q\_{\mathrm{prim}}(s,z^{\*}(s)) - \sup\_{z\neq z^{\*}(s)}Q\_{\mathrm{prim}}(s,z)$$
> satisfies $\Delta_{\mathrm{prim}}(s) > 2\delta_\gamma$, where $\delta_\gamma := \sup_{s,z} V_{\max} \cdot \mathbb{E}\left[ \big|\gamma_{LL}^{\tau}-\gamma\big| \mid s,z \right]$, then the maximizing action is unchanged. So LMTA intentionally optimizes a **decision-time** objective, and the deviation from primitive-time discounting is explicitly controlled.
>
> **3. Sampled MuZero baseline**
>
> We implemented **Sampled MuZero** under matched compute:
>
> | Method         | AIM $(N=500, K=70)$ | SOP $(N=500, K=70)$ |
> | :------------- | ------------------: | ------------------: |
> | Flat MuZero    |       260.60 ± 8.41 |        17.75 ± 0.30 |
> | Sampled MuZero |       283.93 ± 2.55 |        19.41 ± 0.46 |
> | **LMTA**       |   **324.15 ± 2.38** |    **31.60 ± 1.94** |
>
> Sampled MuZero improves over Flat MuZero, but remains substantially below LMTA, suggesting that action sampling helps with primitive-action branching but not the central SSCO challenge of variable-duration abstraction. Learning curves are shown in **Fig. 4 of https://anonymous.4open.science/r/LMTA/ICML_rebuttal.pdf**.
>
> **4. Compute–performance tradeoff**
>
> Table 19 already shows LMTA is more expensive than WS-Option at inference, but much cheaper than flat MuZero-style planning. We therefore added a compute–performance analysis in **Fig. 1 at the anonymous link**, plotting reward against matched wall-clock latency.
>
> The main result is that **LMTA has the strongest reward–latency frontier overall**. It performs strongly even in the greedy (**no-search**) setting, because MCTS provides improved root policy/value targets during training, so much of planning’s benefit is amortized into the learned high-level policy; it improves further with modest planning, and then plateaus. By contrast, **WS-Option** plateaus around **301–302**, and **Sampled MuZero** around **290**. Thus LMTA does **not** require very large test-time search budgets to achieve strong performance.

---

> > ### Author Rebuttal · Reviewer_pWAR · 2026-04-03
> >
> > Thanks for the author's responses and putting together the additional results on short notice. I feel that my questions have been adequately answered, and I agree with the other reviewers for an acceptance.

---

> > > ### Author Response · Authors · 2026-04-03
> > >
> > > Thank you for the updated review and for the thoughtful feedback throughout the review process. We’re glad the rebuttal addressed your concerns.

---

### Official Review · Reviewer_YvBw · 2026-03-02

**Soundness:** 3
**Presentation:** 1
**Significance:** 3
**Originality:** 3
**Overall Recommendation:** 5
**Confidence:** 4

**Summary:**

This paper proposes SSCO, an approach to address issues associated with hierarchical RL (HRL) based on world models to the domain of Sequential Stochastic Combinatorial Optimization (SSCO). The core issue here is the variable duration of high-level actions, making it challenging to use a single predictive model to predict outcomes of both macro (high-level) and micro (primitive) actions. The authors approach this by an approach driven by MuZero (MCTS on a latent states of learned transition model) at the high level of decision-making. This world model is defined in a unit sphere latent space and trained using a composite loss devised for this papers' purposes. The paper evaluates the proposed method on three relevant environments and claims improvements in obtained compared to included SotA learning-based baselines and heuristics.

**Compliance With Llm Reviewing Policy:**

Affirmed.

**Final Justification:**

My original concerns have been addressed, I raised my score to 5. The remaining concern is limited evaluation on non-SSCO problems, making 5 the most suitable score from my perspective.

**Key Questions For Authors:**

1. What is the benefit of having a shared world model for both high-level and low-level control? How specific is this for the SSCO problem? A better motivation for having this single world model would help me appreciate the relevance of the contributions better.
2. How does the approach perform in terms of computational cost compared to the other approaches (lacking in Table 19)?
3. Can you compare the approach to an exact solver on a small-scale problem to assess performance of learning approaches?
4. How is the subgoal space $\mathcal{Z}$ defined? Is it a subspace of the latent space $\mathcal{H}$? If not, where does it come from?
5. Can you provide any theoretical or empirical guidance on how to set some of the hyperparameter values, notably $\kappa$ and the weights for the loss components?
6. Why are not all approaches, comparing Figure 5 and 2, included in the main body results in Figure 2? Specifically director seems to be a painful omissing here?

**Limitations:**

Additional computational costs should be mentioned in the discussion.

**Strengths And Weaknesses:**

*  **Originality/Novelty:** The paper stands in a recent tradition of using HRL for SSCO, adapting this for model-based RL with MuZero could be anticipated yet it is timely. The
  formulation of the subgoal budget prediction using a shared latent space based on a geometrically motivated formulation brings some new concept to this space.
* **Significance:** The problem addressed is highly relevant to the SSCO niche, and potentially for close CO-related niches as well. Being able to set a budget for a low-level policy (in e.g. options framework) may not be that relevant since in most accepted formulations to this setting, the termination is handled within the option rather than by the high-level controller.
* **Quality/Rigor:** Overall the paper is of reasonably high quality, although there seem to be some crucial aspects including details on the subgoal space $\mathcal{Z}$ and the selection of hyperparameters (see questions to authors) where the paper lacks in providing (theoretically or empirically motivated) guidance. In the experimental results I would like to see some results on computational costs as well as a comparison to an exact solver on a deterministic snapshot or via repeated re-optimization, i.e. model predictive control, to assess how close the proposed approach gets to a near-optimum on small-scale problems if this is feasible. The experimental setup is otherwise rigorous within the SSCO subfield and includes most relevant baselines, all relevant ablations, and qualitative assessments. The theoretical results motivating the setup are to the best of my understanding sound and well-constructed.
* **Presentation:** The paper is generally well written and easy to follow but there are some crucial parts where the paper would benefit strongly from an improved presentation, see questions below and detailed remarks. The presentation of the results is dubious, several approaches are missing from the main body results in Figure 2 and the additional computational overhead is not discussed in any of the results or the Discussion section.

### Minor corrections / typos
* Section 3 > Model-based planning with latent space tree search > notation confused me here. $\mathcal{R}$ is defined as the MDP reward function, so shouldn't $g_\theta \dots \to \mathcal{H} \times \mathbb{R}$?  Same for the other mentions of $\mathcal{R}$
* In this same paragraph,  "$g_\theta$ outputs both $(h', R)$" < $R$ seems pooly aligned with previous notation
* Same for "value $v$"
* ln 214, ln 217: $g_\theta$ maps to $\mathcal{R}$ , should this be $\mathbb{R}$?
* eq 1, ln 196, $\pi^L$ has not yet been defined at this point, only from the appendices could I get at its role. The main-body should be self-contained.
* ln 203, ln 207:  $\mathcal{V}\subset\mathbb{R}$ is rather vague, why not simply use $\mathbb{R}$ or another definition based on the reward space instead?
* ln 205 how are the policy prior $\mathbf{p}_k$ and policy prior space $\mathcal{P}$ defined?
* ln 212 footnote 1, ln 221 : $\mathcal{B}$ is not introduced yet
* Section 4.2 align usage of $\triangleq$ and $\coloneqq$
* line 253 equation 2a: what is $\kappa$?
* ln607: equation 3 is not in the main text
* ln 955: the algorithm seems to lack some components, including the inputs, some initialisations
* ln1155: empty Appendix F1 / alignment with tables

---

> ### Author Rebuttal · Authors · 2026-03-30
>
> We thank the reviewer for the careful reading and constructive feedback. We agree that the draft needs clearer presentation, especially around notation, the role of the shared world model, the definition of the subgoal space, and the placement of key empirical results.
>
> **1. Shared world model**
>
> The main benefit is not that LMTA uses the same model for direct online planning at both levels, but that it learns **one latent geometry compatible with both primitive-step smoothness and variable-duration macro transitions**. This matters in SSCO because the high-level policy lives in an SMDP: a chosen subgoal and budget induce a stochastic, variable number of primitive steps before the next high-level decision. If the latent space were shaped only for macro prediction, it could collapse local primitive structure; if shaped only for primitive transitions, it would not support useful one-step macro planning. The shared world model lets the same encoder $h_\theta$ support both. Its **direct planning role at inference is high-level** through $g_\theta^h$ and $R_\theta$, while the low-level policy benefits **indirectly** from the learned latent geometry. We will revise the introduction and Section 4.2 to make this distinction explicit.
>
> **2. Computational cost**
>
> We now add a matched wall-clock compute–performance analysis in **Fig. 1 at [https://anonymous.4open.science/r/LMTA/ICML_rebuttal.pdf](https://anonymous.4open.science/r/LMTA/ICML_rebuttal.pdf)** by varying the main test-time compute knobs of LMTA, WS-Option, and Sampled MuZero, and measuring reward against inference latency. **LMTA has the strongest frontier overall**: it already performs strongly in the greedy (**no-search**) setting, improves further with modest test-time planning, and then largely plateaus. By contrast, **WS-Option** improves only modestly with additional compute and **Sampled MuZero** plateaus at a substantially lower reward level. Over comparable latency ranges, WS-Option plateaus around **301–302** and Sampled MuZero around **290**, while LMTA remains clearly higher. Thus, LMTA is not dependent on very heavy search at inference time, and most of its gain is retained even at low or moderate test-time compute. The strong greedy result is also consistent with training: MCTS provides search-improved targets, so much of the planning benefit is amortized into the learned high-level policy/value.
>
> **3. Exact solver**
>
> We implemented an **exact expectimax solver with memoization** for the full stochastic sequential AIM process on tiny instances ($N=10, T=4, K=4$), where decision nodes choose the next seed and chance nodes enumerate stochastic cascade outcomes over the evolving state. We used this exact stochastic solver, rather than a deterministic snapshot or MPC surrogate, as AIM is inherently a **stochastic sequential decision problem with adaptive decisions and evolving state**.
>
> On these instances, the exact solver gives an average optimal value of **8.91**. **LMTA** achieves **8.73**, corresponding to an average gap of **0.18** and a mean ratio of **0.98** to the exact optimum. **WS-Option** achieves **7.72**, with a gap of **1.19** and a mean ratio of **0.87**. Thus, LMTA is substantially closer to the exact optimum than WS-Option.
>
> **4. Subgoal space $\mathcal{Z}$**
>
> In LMTA, subgoals are **learnable embeddings** discovered end-to-end during training; the learned subgoal count is $|\mathcal{Z}|=32$ in our main setting. The planner's policy head outputs a prior over this discrete subgoal set, and the dynamics model takes a latent state together with a subgoal $z\in\mathcal{Z}$ as input. So $\mathcal{Z}$ is **not** a geometric subspace of the latent state space $\mathcal{H}$; it is a finite learned set of abstract subgoal embeddings that interacts with the latent state through $g_\theta^h(h,z)$, $R_\theta(s,z)$, and $b_\psi(\cdot\mid h,z)$.
>
> **5. Hyperparameters**
>
> The implementation already uses a simple default across AIM and SOP: $w_{\text{cap}}=w_{\text{push}}=w_{\text{order}}=0.10$, with $\kappa=0.10$, $\lambda_m=10^{-4}$, and $|\mathcal{Z}|=32$. The intended guidance is:
> (1) $\kappa$ should be small so one primitive step remains a local latent move;
> (2) the unified-loss weights were set equal to avoid overfitting the geometry to any single force and to reflect the complementary roles of local Pull, macro Push, and duration Order;
> (3) the theory appendix already connects these terms to explicit approximation quantities.
>
> **6. Figure 2 baselines**
>
> In the submitted version, Figure 2 focused on the most established SSCO comparison set centered on WS-Option, while additional baselines such as Director and the MuZero-style methods were reported separately. To make the comparison more complete during rebuttal, we now provide **updated Figure 2 plots that include all evaluated baselines**, including Director and the MuZero-style baselines, under the same evaluation pipeline. These plots are shown in **Fig. 3 at the above anonymous link**.

---

> > ### Author Rebuttal · Reviewer_YvBw · 2026-04-01
> >
> > My questions have mostly been adequately addressed, however the details on the subgoal space $\mathcal{Z}$ have raised a new concern wrt the interpretability/controllability of the agent using the subgoals. E.g. could the subgoals somehow be used in human-in-the-loop deployment, be used to explicitly controlling the agent by selecting or suggesting a subgoal, or be used to explain agent decision making to human observers? Or are these subgoals limited to some learned embedding without clear human-understandable semantics at this stage?
> >
> > My final remaining concerns are related to applicability and evaluations being limited to specifically SSCO, whereas the approach has impact potential in other CO areas as well. Although I invite the authors to clarify their position on the applicability of the approach, I acknowledge that new claims on applicability beyond SSCO and any support for such claims are out of scope, and clarify that I merely pose this question here out of general interest.
> >
> > *edit after rebuttal response*
> > My concerns have now been fully addressed, I will raise my score to 5. The remaining concern is limited evaluation on non-SSCO problems, making 5 the most suitable score from my perspective.

---

> > > ### Author Response · Authors · 2026-04-01
> > >
> > > We thank the reviewer for this thoughtful follow-up. At this stage, the learned subgoals are best viewed as **learned latent abstractions**, so we do not claim that they yet form a fully human-authored control vocabulary. At the same time, they are not arbitrary. **In Figure 3 in the appendix, we visualize the actions induced by conditioning the low-level policy on different learned subgoals on exemplar AIM and SOP graphs.** The resulting behaviors are qualitatively distinct and strategically meaningful: in AIM, one subgoal focuses on central high-degree hubs while another targets a dense local community; in SOP, one subgoal selects scattered high-profit targets while another prefers a spatially dense neighborhood. These examples suggest that the learned subgoals already capture target regions and structural patterns that help the planner assemble coherent long-horizon strategies. We therefore view them as a promising basis for future human-in-the-loop control or explanation, even though they are not yet explicitly labeled in human-interpretable terms.
> > >
> > > On applicability beyond SSCO, our view is that the method is most natural for problems with the same structural pattern: a high-level planner selects a subtask or subgoal, and executing that choice may require a **variable number of low-level interactions** before control returns to the high level. This variable-duration structure is exactly what motivates our SMDP formulation and differentiates LMTA from fixed-stride hierarchical world models. While our experiments focus on SSCO, we therefore expect the underlying idea to be relevant more broadly to other hierarchical decision problems with **resource-constrained, variable-duration temporal abstraction**, including some combinatorial planning settings and some robot-learning settings with long-horizon subtasks.

---

### Official Review · Reviewer_nxgh · 2026-03-13

**Soundness:** 3
**Presentation:** 2
**Significance:** 3
**Originality:** 3
**Overall Recommendation:** 5
**Confidence:** 3

**Summary:**

The paper begins with an introduction of SSCO (Sequential Stochastic Combinatorial Optimization) problems, which  range from logistics to social networks. The paper explains that standard "flat" RL methods often fail in these settings due to the very large number of actions and the necessity to reason over unstructured long horizons. For this reason, hierarchical RL (HRL) is often used to tackle the problem, enabling the decoupling of "sub-goal selection" and "resource budgeting" from the lower level "how do we actually solve this particular sub-goal". However, the high-level HRL agent now operates in a semi-Markov decision process, requiring specialized RL tooling.

# Claimed Contributions

The authors propose a HRL framework called LMTA:

- Model based
- Goal conditioned
- Subgoals are learned
- "Budget head" outputs budget for each subgoal
- The next subgoal is picked using a tree search planner
- Also uses a world model that tackles the higher level transition. In other words, the high-level agent can predict the next high-level state using the  world model.
- The world model also enables micro/low level steps

# Method

The proposed method uses Monte Carlo Tree Search (MCTS) as the high level planner. It plans over a learned world model, which is trained using a loss formulation where low-level actions have a bounded subsequent-distance (similar to a Markovian assumption), high-level transitions for subgoal $z$ are "at least bigger" than some learned margin $m_\phi(z)$ (encourages macro-level to be adequately "long", time-wise), and temporal order encourages latent subgoal representations to take their realized budget from past rollouts into account. This loss formulation is a core part of the paper's contribution.

The high level MCTS planner uses the world model and selects the next subgoal. Then, another learned component (a budget policy) produces a budget allocation for said subgoal. The subgoal is then tackled by the lower hierarchical level.


# Experiments

The experiments are conducted on two problem settings: SOP (Stochastic Orienteering Problem) and AIM (Adaptive Influence Maximization). The appendix also contains results for Power-2500. LMTA shows improvements over baselines in all cases.

An ablation study over LMTA's different components is also provided.

**Compliance With Llm Reviewing Policy:**

Affirmed.

**Final Justification:**

The rebuttal addressed my main concerns by clarifying certain points.

**Key Questions For Authors:**

1. Can the authors clarify how $g_\theta$ connects with claims from earlier in the paper that established LMTA as using a multi-scale, variable-scale, world model? It seems that while the model is indeed trained to respect some low-level markov state distance, and is trained to handle variable-time higher-level dynamics - this capability is only leveraged as part of the high-level planner to obtain better higher-level world modelling. Yet, in the introduction, it seems like it is implied that $g_\theta$ can also be used for immediate low-level planning.

In LMTA, the budget head produces a budget prediction only after the HL policy has selected a subgoal. The decision to perform resource allocation *after* sub-goal selection seems to preclude certain types of reasoning that might be very useful in many domains. For instance, it might be the case that certain tasks are simply not achievable given some (low) remaining budget - and that this should be taken into account at the highest level of planning. Or, in some environments, it might the case that composing subgoals A+B+C result in the same reward as doing subgoal D, and for a lower budget - but because budgeting is not handled by the HL planner, there is no incentive to select A+B+C over D beyond the planner searching for the optimal path. But given that the planner plans over a learned model and that the budget head is also learned, there is no guarantee that the planner actually achieves this, and strategies could be implemented to encourage this. The footnote on page 4 does explain that conditioning on the budget would cause a larger search space.

2. Has LMTA successfully handled cases like the aforementioned "A+B+C vs. D = same reward, different budget"? Is there a way to plan for both budget and subgoal as part of planning? Could the authors clarify how the planner accounts for budgeting?
3. How do the authors view the relationship between budget and optimization? Could future work based on your method treat the budget as an optimization constraint instead (akin to cost in safe RL)?
4. In practice, how is the low-level realized runtime information collected to train the budget policy and world model? Algorithm 1 outlines that the low-level policy is stopped if it goes past the allocated budget, so it would seem that the collected runtime information might not reflect true runtime information.

**Limitations:**

Yes. The impact statement is clear and well formulated. The limitations are also adequate.

**Strengths And Weaknesses:**

The reviewer believes that this is a strong technical contribution that would be of interest to fields realted to HRL and SMDPs as a whole. The loss design for the world model is quite elegant and is of strong interest to hierarchical world modelling. The experiments use a large number of seeds (10), which is appreciated.

## Minor concerns

- The paper can be too verbose at times (method section, into, related work often repeat themselves).
- Some content from the appendix would be better appreciated as part of the main resutls (tables 14, 15, and the Power-2500 results.)
- LMTA is not introduced properly. What does it stand for?
- A diagram figure would have helped to understand the system. As it is, it is hard to tell how each piece interact with each other piece. Figure 1 is on page 4, and is easier to parse once having digested the method section. An easily digestible, stylized, "graphical abstract" type of system diagram would be helpful at the top of page 2.
- Suggest reformating the contributions and rephrasing them more clearly. For instance, the learned dynamics model is first introduced as a high-level world model, and then it is clarified that it also handles low-level actions, which can lead to confusion.
- The paper claims that model-free methods limit forward planning under stochasticity. An alternative point of view is that model-free methods avoid model-based error accumulation/model drift problem, which is not discussed much in the paper. It would have been interesting to have a study on the modelling power of the dynamics function.

## Weaknesses

- The paper's experimental regime is limited to variants of two main tasks (SOP and AIM), as well as another task (Power-2500) which is reported in the appendix.
- The improvement of LMTA over the options baseline in the AIM task is of ~8%

---

> ### Author Rebuttal · Authors · 2026-03-30
>
> We thank the reviewer for the careful reading and helpful suggestions.
>
> **1. The role of the multi-scale world model**
>
> In LMTA ("Learning Multi-Timescale Abstractions"), the multi-scale / variable-scale property is primarily a **representation-learning property** of the shared latent space: the latent space is trained to respect both primitive-step transitions and variable-duration macro transitions. The world model's **direct role at inference time is at the high level**: the planner uses it to predict the post-subgoal latent state and macro reward in one step. The low-level controller does **not** run model rollouts online; it is a subgoal-conditioned action-value policy that benefits **indirectly** from the learned latent geometry and subgoal embeddings. We will revise the introduction and method sections to make this distinction explicit.
>
> **2. Why budget is chosen after subgoal selection and how the planner still accounts for budgeting**
>
> This is a deliberate tractability choice. In LMTA, the planner searches over subgoals $z$, and the budget is then chosen as a second-stage, subgoal-conditioned decision via $b\_\psi(\cdot\mid h\_k,z\_k)$. Explicitly branching over both subgoals and budgets inside MCTS would enlarge the search space from $|\mathcal{Z}|$ to $|\mathcal{Z}|\cdot|\mathcal{B}|$, which is costly in SSCO because budgets are discrete and state-dependent.
>
> Budget is still accounted for. The high-level dynamics model is trained to predict the **budget-marginalized** SMDP transition induced by the learned budget policy and low-level controller:
>
> $$P^{\pi^L,b\_\psi}(s\_{k+1},R\_k\mid s\_k,z\_k):=\mathbb{E}\_{b\_k\sim b\_\psi(\cdot\mid h\_k,z\_k)}\big[P(s\_{k+1},R\_k\mid s\_k,(z\_k,b\_k),\pi^L)\big].$$
>
> Thus $g\_\theta(h\_k,z\_k)$ and $R\_\theta(s\_k,z\_k)$ model the post-subgoal latent state and return under this endogenous budget rule, so the planner accounts for budgeting **implicitly through the learned budget-marginalized transition/value model**, rather than by explicitly branching over budgets.
>
> This is also consistent with Table 1: moving from variant B to C, where budget becomes subgoal-conditioned, improves performance in both AIM and SOP. For the reviewer's (A+B+C) vs. (D) example, the current method handles this tradeoff implicitly rather than through explicit joint $(z,b)$ search. We agree that explicit joint planning over both could model such tradeoffs more directly, and view that as a promising future direction.
>
> We also tested a budget-aware search variant by inserting budget sampling into MCTS. This chance-budget MCTS variant produced lower mean reward and higher variability than default LMTA: on AIM, **324.15 ± 2.38** for LMTA versus **322.8 ± 2.93** (3 samples) and **321.9 ± 3.12** (1 sample); on SOP, **31.60 ± 1.94** versus **29.5 ± 2.31** and **29.1 ± 2.45**. The degradation is larger with fewer budget samples because the search then relies on a noisier Monte Carlo estimate of the budget-marginalized transition/value. These results support our design choice: in the SSCO settings we study, it is more effective to keep MCTS focused on subgoal lookahead and handle budget as a learned second-stage decision.
>
> **3. Relationship between budget and optimization / constrained formulations**
>
> In the current paper, budget is treated as a **decision variable** allocated after subgoal selection. More broadly, we agree there is a natural connection to constrained optimization and CMDP-style formulations, where budget could instead be treated as an explicit cost or constraint. Extending LMTA toward explicit constrained planning is an interesting future direction.
>
> **4. How realized runtime information is collected**
>
> The training target is the **realized executed duration** of the macro-action under the deployed hierarchical controller, not a counterfactual "full completion time." In both AIM and SOP, at each high-level decision the low-level controller selects primitive actions up to the allocated budget $b\_t$, and execution may terminate earlier if no legal actions remain; thus the realized duration is a stopping time $\tau(s,z,b)\le b$.
>
> So the signal is not intended to estimate an unconstrained runtime. It is simply the duration of the **executed** SMDP transition induced by the chosen subgoal, budget, low-level policy, and stopping rule, exactly the quantity needed by our budget-aware SMDP formulation and the pairwise duration-ordering objective in Appendix A.
>
> **5. Model-based planning versus model error**
>
> We agree that the discussion of model-free vs. model-based methods should be more balanced. Our intention was not to claim that model-free methods are uniformly worse; rather, model-based methods enable explicit forward planning under stochasticity while also introducing the usual risk of model error accumulation. We will revise the discussion accordingly.

---

> > ### Author Rebuttal · Reviewer_nxgh · 2026-04-02
> >
> > Thank you for the clarifying answers.
> >
> > Regarding Q4; it would seem that if an execution can only decrease the reported budget cost (because of the hard stopping rule), there might be some sort of collapsing behaviour. The author's results are, however, evidence that this is not a big issue. But it might be worth a short discussion section in the final version.
> >
> > I have updated my score to 5.

---

> > > ### Author Response · Authors · 2026-04-03
> > >
> > > Thank you for the updated review and for the helpful feedback. We’re glad the rebuttal addressed your concerns. In the final revision, we will add a short discussion of this point.

---

### Official Review · Reviewer_W55E · 2026-03-13

**Soundness:** 2
**Presentation:** 3
**Significance:** 3
**Originality:** 4
**Overall Recommendation:** 5
**Confidence:** 4

**Summary:**

The paper introduces LMTA, an algorithm that solves the sequential stochastic combinatorial optimization problem using a latent-space tree search planner to search over learned subgoals and then choose a budget conditioned on the subgoal. In order to incorporate both primitive and macro transitions in the same world model, LMTA also learns a multi-timescale SMDP world model which encode the latent states on a unit sphere so that the temporal scale is encoded in the geometry of the latent space. This allows the LMTA to shape the latent space while respecting both high and low level transitions. LMTA demonstrates significant improvements over previous baselines on AIM, SOP and Power-2500 tasks.

**Compliance With Llm Reviewing Policy:**

Affirmed.

**Final Justification:**

The rebuttal addressed my main concerns and further strengthened the paper.

**Key Questions For Authors:**

1. Have the baselines been hyperparameter tuned?
2. How does LMTA perform on the original problem size from Feng et al?
3. Can the computational cost of LMTA be reduced? How does it perform compared to WS-Option on similar compute budget?

**Limitations:**

yes

**Strengths And Weaknesses:**

Soundness:
Strengths: The paper's claims are well supported by consistent improvements over baselines across multiple configurations, a progressive ablation study that isolates each component's contribution, and per-loss-term ablations that validate the multi-timescale objective design. The theoretical analysis provides a clean chain of results connecting the training objective to planning regret.
Weaknesses: The paper scales up environments from Feng et al. (2025) to N=500 nodes but does not justify why this scale was chosen or analyze how each algorithm's performance degrades as problem size increases. A scaling curve showing performance versus node count would clarify whether LMTA's advantage comes from better asymptotic scaling or a constant offset. More importantly, it is unclear whether baselines were independently hyperparameter-tuned for the larger instances. If LMTA received careful tuning (as evidenced by the sensitivity sweeps in Appendix I) while baselines used transferred or default hyperparameters, the comparison may be inflated. Including results on the original node sizes from Feng et al. would provide a more direct and verifiable comparison point.

Presentation:
The overall writing quality is strong and the method is clearly motivated. However, the main quantitative results (Tables 14 and 15) are moved to the appendix, forcing readers to leave the main text to evaluate the core claims. Relocating parts of Section 5.2 or Figure 2 to the appendix would free space to include these tables in the main body.

Significance:
Strengths: The paper opens a promising research direction by demonstrating that SMDP-aware multi-timescale world models can substantially improve planning in sequential combinatorial domains.
Weaknesses: It is clear that this approach is significantly more computationally costly compared to previous work (WS-Option). From the provided information, it would seem that LMTA utilizes >4x (400%) more time but only improves by (7%). This can significantly impact its practicality. The paper does not discuss whether this tradeoff is favorable in practice or whether there are ways to reduce LMTA's overhead.  Additionally, the computational cost is measured in wall-clock time which is not a standardized measure of computational cost. Reporting FLOPs or environment interaction counts alongside wall-clock time would provide more reproducible cost measures. Additionally, a discussion on the scalability of each algorithm might help. Can MCTS simulation count be reduced with minimal performance loss (the sensitivity sweep in Table 6 suggests it can, since 50 simulations performs nearly as well as 150)? Could WS-Option's performance improve with additional compute? A discussion of the compute-performance Pareto frontier for each method would strengthen the practical case for LMTA.

Minor: A visualization of the latent space such as T-SNE might improve the paper, but is not required.

Originality:
The paper makes strong novel contributions. The combination of MuZero-style tree search with an SMDP-aware world model for combinatorial optimization is new, and the multi-timescale latent geometry objective is a creative alternative to explicit duration prediction. The subgoal-conditioned budget allocation, while building on actor-critic machinery from prior work, is a meaningful architectural contribution that clearly separates strategic intent from resource allocation.

---

> ### Author Rebuttal · Authors · 2026-03-30
>
> We thank the reviewer for highlighting three issues that are especially important for strengthening the paper: fairness of the large-scale comparison, direct comparability to the original Feng et al. setting, and the practical compute–performance tradeoff.
>
> **1. Have the baselines been hyperparameter tuned?**
>
> Yes. For the prior SSCO baselines from Feng et al., especially WS-Option and the domain heuristics, we re-checked the large-instance settings and swept the main performance-critical knobs on the larger AIM/SOP instances. Flat DQN, Director, and MuZero-style methods were additional baselines that we reimplemented/adapted and tuned for SSCO under the same evaluation pipeline as LMTA, since such baselines were not previously available in this domain.
>
> **2. How does LMTA perform on the original problem size from Feng et al.?**
>
> We added AIM results at the original Feng et al. graph size, **N = 200**, under the same four benchmark settings from their Table 1: $(T,K)\in{(10,10),(10,20),(10,30),(20,10)}$.
>
> | Method    |      $T=10,K=10$ |       $T=10,K=20$ |       $T=10,K=30$ |      $T=20,K=10$ |
> | :-------- | ---------------: | ----------------: | ----------------: | ---------------: |
> | WS-Option |     78.84 ± 2.86 |     116.21 ± 3.71 |     128.63 ± 3.89 |     81.14 ± 2.93 |
> | **LMTA**  | **83.21 ± 2.41** | **123.83 ± 3.12** | **137.28 ± 3.46** | **86.16 ± 2.66** |
>
> LMTA improves over WS-Option at the original benchmark size across all four settings, so the gain is not an artifact of scaling only to $N=500$. To address the broader scaling concern, we also added a same-protocol AIM scaling curve with fixed $T=10, K=30$ and $N\in{200,300,400,500}$ in **Fig. 2 at [https://anonymous.4open.science/r/LMTA/ICML_rebuttal.pdf](https://anonymous.4open.science/r/LMTA/ICML_rebuttal.pdf)**, retraining and evaluating both LMTA and WS-Option at each size. The gap is already present at $N=200$ and remains clear as $N$ increases.
>
> In addition, to calibrate performance against the exact optimal expected return, we conducted a tiny-scale exact-solver study for AIM ($N=10, T=4, K=4$). LMTA achieves 8.73 versus an exact optimum of 8.91 (mean ratio 0.98), whereas WS-Option achieves 7.72 (mean ratio 0.87), showing that LMTA is substantially closer to the exact optimum when exact solution is tractable.
>
> **3. Computational cost / practical tradeoff**
>
> We added a **matched wall-clock compute frontier** in **Fig. 1 at [https://anonymous.4open.science/r/LMTA/ICML_rebuttal.pdf](https://anonymous.4open.science/r/LMTA/ICML_rebuttal.pdf)** that varies the main **test-time planning budget** of both **LMTA** and **WS-Option** and reports reward against inference latency.
>
> The main result is that **LMTA remains the strongest method across the entire wall-clock frontier**. LMTA already achieves strong reward in the **greedy (no-search)** setting (**321.8**), improves with additional planning, and then **largely plateaus** around **326.2**. This small greedy-to-full gap shows that LMTA is **not highly dependent on heavy test-time search**. It is also consistent with how LMTA is trained: even when online search is removed at test time, the high-level policy/value are still trained with **search-improved targets**, so much of the planning benefit is already amortized into the learned policy. WS-Option also improves with more compute, from about **292.5** to about **302.0**, but starts much lower and still plateaus well below LMTA. Thus, the practical conclusion is not that LMTA gains more from extra compute, but that **LMTA starts from a much stronger low-compute operating point and remains better throughout the tested compute range**.
>
> We also report **interaction-count measures inline**. On this AIM benchmark ($T=10$, budget $=70$), real environment interaction is essentially matched across methods: all methods use about **10 high-level decisions** and about **69–70 low-level actions / seeds per instance**. The main difference therefore comes from **internal planner compute** rather than extra environment interaction.
>
> Overall, LMTA is not universally cheap, but its **compute–performance tradeoff is substantially better than the original single-point comparison implied**: it performs strongly even with little or no online search, benefits from modest additional planning, and then shows diminishing returns rather than depending on very large test-time compute.
>
> We also agree that the main quantitative tables should be reported more prominently. In the revision we will move the key AIM/SOP comparison tables from the appendix into the main text and shift lower-priority material to the appendix.

---

> > ### Author Rebuttal · Reviewer_W55E · 2026-04-02
> >
> > My concerns have been resolved. In the final revision, I would like to see the details on hyperparameter tuning of prior baselines. Additionally, my original concern regarding reporting and measuring compute complexity in wall-clock time remains. But overall, I am satisfied with the additions and will update my score to 5.

---

> > > ### Author Response · Authors · 2026-04-02
> > >
> > > Thank you for the updated review, as well as the helpful feedback to improve the final revision. We’re glad the rebuttal addressed your main concerns. In the final revision, we will clarify how the prior baselines were tuned and further improve the discussion of compute tradeoffs.

---

### Decision · Program_Chairs · 2026-04-30

**Decision:**

Accept (regular)

**Comment:**

The reviewers uniformly agreed that the paper makes a strong contribution in proposing a novel approach to learning solutions to hierarchical combinatorial planning problems.

All reviewers had minor suggestions for improving the paper and the authors should take those into account when finalizing the camera ready version.